# Boosting Neural Combinatorial Optimization for Large-Scale Vehicle Routing Problems

**Fu Luo**[1], **Xi Lin**[2], **Yaoxin Wu**[3], **Zhenkun Wang**[1,*], **Tong Xialiang**[4], **Mingxuan Yuan**[4],
**Qingfu Zhang**[2]

[1]Southern University of Science and Technology
[2]City University of Hong Kong
[3]Eindhoven University of Technology
[4]Huawei Noah's Ark Lab

luof2023@mail.sustech.edu.cn, xi.lin@my.cityu.edu.hk,
y.wu2@tue.nl, wangzhenkun90@gmail.com, tongxialiang@huawei.com,
yuan.mingxuan@huawei.com, qingfu.zhang@cityu.edu.hk

## Abstract

Neural Combinatorial Optimization (NCO) methods have exhibited promising performance in solving Vehicle Routing Problems (VRPs). However, most NCO methods rely on the conventional self-attention mechanism that induces excessive computational complexity, thereby struggling to contend with large-scale VRPs and hindering their practical applicability. In this paper, we propose a lightweight cross-attention mechanism with linear complexity, by which a Transformer network is developed to learn efficient and favorable solutions for large-scale VRPs. We also propose a Self-Improved Training (SIT) algorithm that enables direct model training on large-scale VRP instances, bypassing extensive computational overhead for attaining labels. By iterating solution reconstruction, the Transformer network itself can generate improved partial solutions as pseudo-labels to guide the model training. Experimental results on the Travelling Salesman Problem (TSP) and the Capacitated Vehicle Routing Problem (CVRP) with up to 100K nodes indicate that our method consistently achieves superior performance for synthetic and real-world benchmarks, significantly boosting the scalability of NCO methods. The code is available at https://github.com/CIAM-Group/SIL.

## 1 Introduction

The vehicle routing problem (VRP) is a typical type of combinatorial optimization problem (COP) and is often encountered in numerous real-world applications (Garaix et al., 2010; Brophy & Voigt, 2014; Elgarej et al., 2021). Due to the NP-hard nature, solving VRPs remains extremely challenging (Ausiello et al., 2012). Traditional methods are generally hindered by their heavy reliance on domain expertise and tuning work in algorithm design. Meanwhile, they often suffer from low computational efficiency that hampers their applicability on large-scale VRP instances.

Recently, the neural combinatorial optimization (NCO) methods for solving VRPs in an end-to-end manner have attracted considerable attention (Bengio et al., 2021). These methods build deep neural models to automatically learn problem-solving policies from data, significantly mitigating the need for costly manual effort in algorithm design. The learned policy can efficiently generate approximate solutions for VRP instances. NCO methods have gained comparable or even superior performance to traditional methods on small-scale problem instances with no more than 100 nodes (Kool et al., 2019; Kwon et al., 2020; Hottung et al., 2022), especially on Traveling Salesman Problem (TSP) and Capacitated Vehicle Routing Problem (CVRP) instances.

Nevertheless, existing NCO methods often struggle when applied to large-scale VRPs. Some efforts have been devoted to training neural models on larger VRPs with up to 500 nodes (Jin et al., 2023; Zhou et al., 2023), aiming to enhance their generalization for solving large-scale VRPs. However, the

---

*corresponding author

difficulty of training will increase drastically as the size of the problem grows, resulting in the inability to obtain sufficient generalization capabilities. Consequently, some methods resort to simplifying large-scale VRPs via decomposition or learning local policies (Pan et al., 2023; Ye et al., 2024; Gao et al., 2024; Fang et al., 2024). The decomposition-based subproblem solver can be trained by learning to construct either a complete solution of a small-scale VRP or partial solutions (e.g., some segments of a TSP solution) of a large-scale one (Kim et al., 2021; Cheng et al., 2023; Luo et al., 2023). The method based on local policy reduces the decision space into the current node's neighborhood in each construction step.

Despite the above efforts in solving large-scale VRPs, current NCO methods still suffer from two obstacles in terms of scalability. First, they usually rely on the conventional self-attention mechanism with high computational complexity (Vaswani et al., 2017), which severely restricts the model's efficiency in constructing a complete solution or multiple partial solutions with large sizes for a large-scale VRP. Second, their models are often trained in supervised learning (SL) or reinforcement learning (RL) manner, both of which grapple with effective training on large-scale VRPs. On the one hand, the SL-based NCO has difficulty in obtaining sufficient (near-)optimal solutions as labels; on the other hand, the RL-based model training suffers from severe sparse rewards as well as high GPU memory usage.

In this paper, we propose a lightweight cross-attention mechanism with linear computational complexity, which can significantly improve the efficiency of the NCO model in solving large-scale VRPs. Unlike the conventional self-attention that makes each node attend to all the other nodes in an instance, the cross-attention reforms the computational process through representative nodes. In particular, the representative nodes first attend to each node of an instance to update their own embeddings, and then the instance nodes' embeddings are updated via attending to the representative nodes. With a fixed number of representative nodes, the computational complexity is greatly reduced in comparison to the conventional self-attention while maintaining effective attention computation between nodes. Based on the proposed cross-attention mechanism, we develop a novel Transformer network for solving large-scale VRPs more efficiently. In addition, we propose an innovative Self-Improved Training (SIT) algorithm that empowers our model to be successfully trained on large-scale instances. The SIT employs the Transformer network itself to refine the solution via iterative reconstruction. The improved solutions, in turn, serve as pseudo-labels to train the network. By iterating solution reconstruction and network training, the SIT enables our NCO method to effectively solve large-scale VRP instances without any labeled data.

We conduct comprehensive experiments on both synthetic and real-world TSP and CVRP benchmarks. The results demonstrate that our NCO method achieves state-of-the-art performance on large-scale VRPs with up to 100K nodes. Our ablation study reveals the effect of the cross-attention and SIT algorithm in improving computational efficiency and solving performance for large-scale VRPs.

## 2 RELATED WORK

### 2.1 GENERALIZATION-BASED METHOD

The generalization-based methods usually train the neural models on small-scale instances and then test them on the same-scale or larger-scale instances. It generally refers to the construction-based method that learns a model to construct approximate solutions for given problem instances in an autoregressive manner. Pioneering works (Vinyals et al., 2015; Bello et al., 2016; Nazari et al., 2018) show that neural models trained with supervised learning (SL) or reinforcement learning (RL) can achieve promising results on small-scale CO problems. Kool et al. (2019) and Deudon et al. (2018) leverage the Transformer structure Vaswani et al. (2017) to develop powerful attention-based models to solve small-scale VRPs. Since then, various Transformer-based methods have been proposed with different improvements (Xin et al., 2021a; 2020; Kwon et al., 2020; Hottung et al., 2022; Kim et al., 2021; Choo et al., 2022; Manchanda et al., 2022). Subsequently, many studies attempt to enhance the performance of neural models on large-scale VRPs (Son et al., 2023; Zhou et al., 2023; Drakulic et al., 2023; Luo et al., 2023). Among them, Drakulic et al. (2023); Luo et al. (2023) employ SL to train the model on 100-node and equip the model with good generalization ability on VRPs with up to 1K nodes. BQ reformulates the Markov Decision Process (MDP) of solution construction to effectively leverage common symmetries of COPs, while LEHD proposes a light encoder and heavy decoder structure to achieve the same goal. However, since the distribution of 100-node instances

differs drastically from that of instances with more than 10K nodes, the features learned from such small-scale instances are not applicable to very large-scale instances, resulting in poor performance on instances with more than 10K nodes. Recent attempts propose to train models on larger-scale instances with up to 500 nodes (Jin et al., 2023; Zhou et al., 2024; Wang et al., 2024; Zhou et al., 2023). However, the training difficulty increases dramatically as the problem sizer grows, resulting in the inability to obtain sufficient generalization capabilities.

## 2.2 SIMPLIFICATION-BASED METHOD

Some methods resort to simplifying large-scale VRPs via decomposition or learning local policies. On the one hand, decomposition-based methods generally transform a large-scale problem into multiple simpler small-scale subproblems, solve them, and then merge their solutions to construct the complete solution of the original large-scale problem. These methods propose different strategies to learn individual models for problem decomposition and subproblem-solving, respectively (Li et al., 2021; Zong et al., 2022; Hou et al., 2023; Pan et al., 2023; Ye et al., 2024). The decomposition-based subproblem solver can be trained by learning to construct either a complete solution of a small-scale VRP or partial solutions (e.g., some segments of a TSP solution) of a large-scale one (Kim et al., 2021; Cheng et al., 2023; Luo et al., 2023). On the other hand, the local policy-based method reduces the decision space into the current node's neighborhood in each construction step. While Gao et al. (2024) adopt an auxiliary policy to bias the model to make decisions within the current node's neighborhood, Fang et al. (2024) directly restricts the decisions to the neighborhood.

## 2.3 HEATMAP-BASED METHOD

In addition to the above works, some heatmap-based methods are proposed to address large-scale TSP instances (Fu et al., 2021; Qiu et al., 2022; Li et al., 2023; Min et al., 2023; Sun & Yang, 2023). This kind of method first builds a graph neural network (GNN) model to predict a heatmap that measures the probability for each edge to be in the optimal solution, and then iteratively searches for an approximate solution using the heatmap (Joshi et al., 2019). Since they rely on the search strategy (e.g., MCTS (Fu et al., 2021)) specifically designed for TSP, they cannot be applied to solve other complicated CO problems such as CVRP. In this work, we primarily focus on the construction-based method without requiring expert knowledge.

## 3 PRELIMINARIES

**VRP Definition.** A VRP instance $S$ can be represented by a graph $\mathcal{G} = (\mathcal{V}, \mathcal{E})$, where $\mathcal{V} = \{v_i\}_{i=0}^{n}$ denotes the node set and $\mathcal{E} = \{(v_i, v_j) | v_i, v_j \in \mathcal{V}, v_i \neq v_j\}$ denotes the edge set. In particular, $v_0$ denotes the depot in some problems (e.g., CVRP).

On the VRP graph, each node is featured by a vector $\mathbf{x}_i \in \mathbb{R}^{d_x}$, with the elements including the node coordinates and other problem-specific attributes (e.g., demands $\{\delta_i\}_{i=0}^{n}$ in CVRP). A solution to a VRP instance is a tour $\boldsymbol{\pi}$, which is a permutation of the nodes. Given a cost function $c(\cdot)$, solving the VRP instance is to search the tour with minimal cost, i.e., $\boldsymbol{\pi}^* = \arg\min_{\boldsymbol{\pi} \in \Omega} c(\pi | \mathcal{G})$, from the feasible tour set $\Omega$. Specifically, the cost function of TSP and CVRP is defined as the Euclidean length of the tour in this work.

A VRP solution is feasible if it adheres to problem-specific constraints, e.g., a feasible solution to a TSP instance is a tour that visits each node in the graph exactly once. Constraints in the CVRP further entail the limited capacity of a vehicle, which are detailed in Appendix B.

**Solution Construction for VRP.** Most NCO methods adopt encoder-decoder-based neural networks to learn the solution construction. The encoder produces node embedding $\mathbf{h}_i$ for each node $v_i$. With node embeddings $\{\mathbf{h}_i\}_{i=0}^{n}$, the decoder sequentially constructs the solution by appending one node to the partial solution at each step. In particular, if none of the nodes have been visited, the partial solution is empty. At each construction step, one node is selected from the unvisited nodes to be added to the partial solution and marked as visited. For example, a partial solution at step $t$ can be represented by $(\pi_1, \pi_2, \ldots, \pi_{t-1})$, where $\pi_1, \pi_{t-1} \in \mathcal{V}$ are the **first** and **last** visited node. The process continues until all nodes are visited and the complete solution is returned.

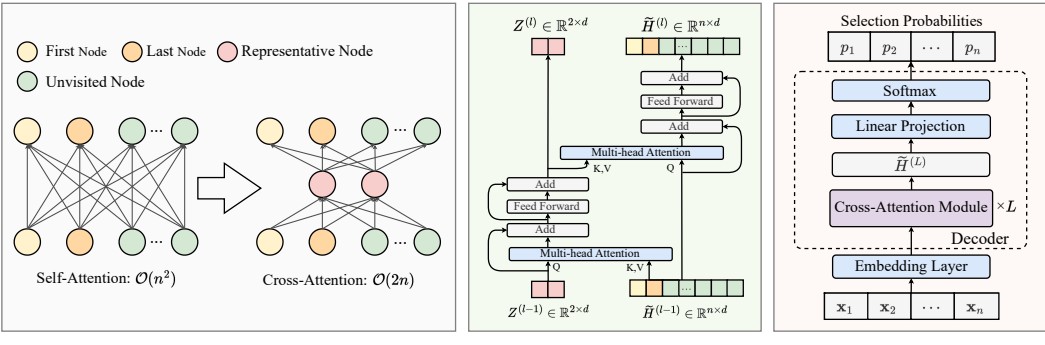

(a) Idea of the Proposed Cross-Attention.    (b) Cross-Attention Module.    (c) Transformer Network.

Figure 1: Cross-attention-based Transformer network. Given less complexity of the cross-attention in (a), we employ it to design the cross-attention module in (b), which utilizes a certain number of representative nodes to lighten the attention in the Transformer network in (c). The resultant Transformer network significantly reduces the amount of computations between nodes, potentially achieving more efficiency in solving large-scale VRPs than current self-attention-based Transformers.

**(Self-)Attention Mechanism.** Given the embedding matrices $X \in \mathbb{R}^{n \times d}$ and $C \in \mathbb{R}^{m \times d}$, where $d$ is the dimension of the embeddings, the scaled dot-product attention can be formulated as:

$$\hat{X} = \text{Attn}(X, C) = \text{softmax}\left(\frac{XW_Q(CW_K)^\mathsf{T}}{\sqrt{d}}\right) \cdot CW_V, \tag{1}$$

where $W_Q, W_K, W_V \in \mathbb{R}^{d \times d}$ are learnable matrices. The attention $\text{Attn}(\cdot, \cdot)$ can aggregate the information from $C$ to $X$. The commonly used Multi-Head Attention (MHA) further performs $h$ attention computations in parallel. We omit $h$ for simplicity throughout this paper. In general, the most complexity of the conventional Transformer architectures originates from calculating the attention in Eq.(1) with $\mathcal{O}(nm)$ computational and memory complexities. Especially if the embedding matrices are identical (i.e., $X = C$), Eq.(1) defines the self-attention with $\mathcal{O}(n^2)$ complexities.

## 4 METHODOLOGY

In this section, we first introduce the cross-attention mechanism tailored for solving VRPs. Then, we provide the cross-attention-based Transformer network and the SIT algorithm. Without loss of generality, we present our method by taking TSP as an example. The implementation details for CVRP are described in Appendix B.

### 4.1 LIGHTWEIGHT CROSS-ATTENTION FOR VRPS

As shown in Figure 1(a), the self-attention (stacked in the encoder or decoder) enforces each input node to interact comprehensively with all the other nodes of a VRP instance for updating their embeddings (Kwon et al., 2020; Pirnay & Grimm, 2024; Luo et al., 2023). However, it inevitably results in $\mathcal{O}(n^2)$ computational and memory complexities, where $n$ is the number of nodes in an instance, i.e., the problem size. As the size increases, these complexities increase drastically, making the model can be hardly trained on large-scale VRP instances. Our empirical results on representative NCO methods verify their quadratic complexities, as shown in Table 4 in the experiments.

To address this issue, we design a lightweight cross-attention mechanism that significantly reduces the computational and memory complexities. As shown in Figure 1(a), we use two representative nodes in the attention calculation. The two representative nodes are first updating their embeddings based on the attention between them and all nodes of the instance. Subsequently, the node embeddings of the instance are updated by conducting attention calculations on the representative nodes. Compared to self-attention, this cross-attention owns a $\mathcal{O}(nm)$ complexity, where $m$ is the number of representative nodes. Thanks to propagating node embeddings through representative nodes, the cross-attention mechanism maintains effective interactions between nodes while achieving low complexity.

## 4.2 TRANSFORMER NETWORK

We develop a cross-attention-based Transformer network for solving large-scale VRPs, following the heavy decoder paradigm in (Drakulic et al., 2023; Luo et al., 2023). As shown in Figure 1(c), the network consists of a single embedding layer and a decoder with $L$ stacked cross-attention modules, which are delineated in the following.

**Embedding Layer.** Given a VRP instance $S$ with $n$ nodes, the embedding layer transforms node features $\{\mathbf{x}_i\}_{i=1}^n \in \mathbb{R}^{n \times d_x}$ to initial node embeddings $\{\mathbf{h}_i\}_{i=1}^n \in \mathbb{R}^{n \times d}$ by a linear projection such that $\mathbf{h}_i = W^{(0)}\mathbf{x}_i + \mathbf{b}^{(0)}, \forall i \in \{1, \ldots, n\}$, where $W^{(0)} \in \mathbb{R}^{d_x \times d}$ and $\mathbf{b}^{(0)} \in \mathbb{R}^d$ are learnable.

**Decoder with Cross-Attention Modules.** At the $t$-th decoding step, we select the first and last nodes $\pi_1, \pi_{t-1}$ from the current partial solution $(\pi_1, \ldots, \pi_{t-1})$ as the representative nodes. These representative nodes reflect the dynamics in the solution construction process, which are commonly used as the context nodes in the literature (Kwon et al., 2020; Luo et al., 2023; Kool et al., 2019). We also explore other configurations of representative nodes and assess their impact on performance. More details can be found in Appendix C.

Given the representative nodes, we design the cross-attention module to advance the node embeddings, as depicted in Figure 1(b). At the $t$-th decoding step, the embeddings of the first and last nodes are denoted by $\mathbf{h}_{\pi_1}$ and $\mathbf{h}_{\pi_{t-1}}$, respectively. The embeddings of unvisited nodes are denoted by $H_a^t = \{\mathbf{h}_i | i \in \{1, \ldots, n\} \backslash \{\pi_{1:t-1}\}\}$. At the first decoding step, one node $\pi_1$ is randomly selected to be the partial solution, and we view it as both the first and last nodes for the next decoding. All the other nodes remain unvisited, i.e., $H_a^0 = \{\mathbf{h}_i | i \in \{1, \ldots, n\} \backslash \pi_1\}$. Accordingly, we define the initial representative node embeddings $Z^{(0)}$ and graph node embeddings $\widetilde{H}^{(0)}$ at the $t$-th decoding step as

$$
\begin{aligned}
Z^{(0)} &= [\mathbf{h}_{\pi_1}W_1, \ \mathbf{h}_{\pi_{t-1}}W_2], \\
\widetilde{H}^{(0)} &= [\mathbf{h}_{\pi_1}W_1, \ \mathbf{h}_{\pi_{t-1}}W_2, \ H_a^t],
\end{aligned}
\tag{2}
$$

where $[\cdot, \cdot]$ means the vertical concatenation operator, and $W_1, W_2 \in \mathbb{R}^{d \times d}$ are learnable matrices. Next, $Z^{(0)} \in \mathbb{R}^{2 \times d}$ and $\widetilde{H}^{(0)} \in \mathbb{R}^{N \times d}$ are processed by $L$ cross-attention modules. In the $l$-th cross-attention module, we first apply representative nodes to attend to the graph node embeddings $\widetilde{H}^{(l-1)}$ to update the representative node embeddings, i.e.,

$$
\begin{aligned}
\hat{Z}^{(l)} &= \text{Attn}(Z^{(l-1)}, \widetilde{H}^{(l-1)}) + Z^{(l-1)}, \\
Z^{(l)} &= \text{FF}(\hat{Z}^{(l)}) + \hat{Z}^{(l)},
\end{aligned}
\tag{3}
$$

where FF denotes the feed-forward layer that is formulated as $\text{FF}(X) = \max(0, XW_{f1} + \mathbf{b}_{f1})W_{f2} + \mathbf{b}_{f2}$, where $W_{f1} \in \mathbb{R}^{d \times d_f}, \mathbf{b}_{f1} \in \mathbb{R}^{d_f}, W_{f2} \in \mathbb{R}^{d_f \times d}, \mathbf{b}_{f2} \in \mathbb{R}^d$ are learnable matrices, and $d_f$ is the hidden dimension of the layer. Subsequently, the graph node embeddings $\widetilde{H}^{(l-1)}$ attend to the representative nodes for updating their embeddings, i.e.,

$$
\begin{aligned}
\hat{H}^{(l)} &= \text{Attn}(\widetilde{H}^{(l-1)}, Z^{(l)}) + \widetilde{H}^{(l-1)}, \\
\widetilde{H}^{(l)} &= \text{FF}(\hat{H}^{(l)}) + \hat{H}^{(l)}.
\end{aligned}
\tag{4}
$$

After $L$ cross-attention modules, the output $\widetilde{H}^{(L)}$ include the advanced embeddings of the first, last, and unvisited nodes. A linear projection and softmax function are applied to embeddings of unvisited nodes $\{\widetilde{\mathbf{h}}_i^{(L)} | i \in \{1, \ldots, n\} \backslash \{\pi_{1:t-1}\}\}$ in order to produce probabilities of selecting each unvisited node, i.e.,

$$
\begin{aligned}
u_i &= \begin{cases} \widetilde{\mathbf{h}}_i^{(L)} W_O, & i \notin \{\pi_{1:t-1}\} \\ -\infty, & \text{otherwise} \end{cases}, \\
\mathbf{p} &= \text{softmax}(\mathbf{u}),
\end{aligned}
\tag{5}
$$

where $W_O \in \mathbb{R}^{d \times 1}$ is a learnable matrix. Each $p_i \in \mathbf{p}$ corresponds to the probability of selecting the unvisited node $i$. We sample the node by the probabilities and add it to the partial solution. A complete VRP solution $\pi = (\pi_1, \ldots, \pi_n)$ is constructed with $n$ decoding steps.

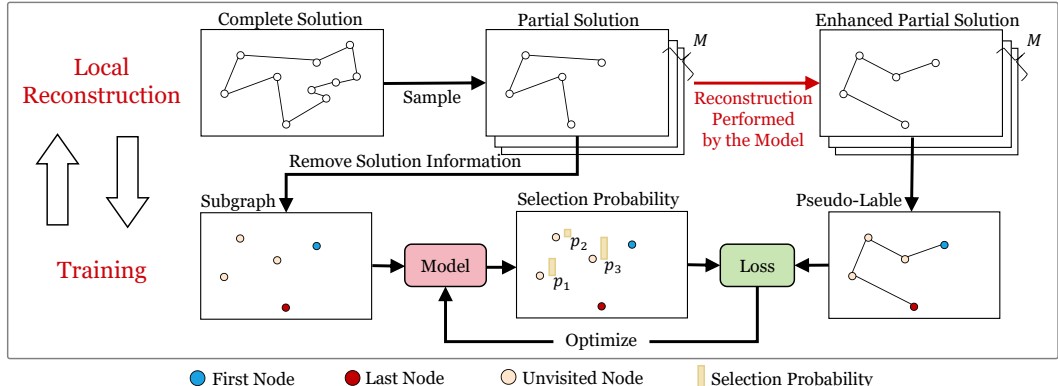

Figure 2: The self-improved training process. In each iteration, the neural model performs multiple local reconstructions (in parallel) to improve the solution quality, and then, the enhanced partial solutions as pseudo-labels are used to train the model to improve its performance.

**Complexity Analysis.** According to Eq.(3), the dimensions of input to cross-attention modules are $\mathbb{R}^{2 \times d}$ for $Z^{(l-1)}$ and $\mathbb{R}^{\widetilde{n} \times d}$ for $\widetilde{H}^{(l-1)}$, where $\widetilde{n} \leq (n + 1)$ is the number of node embeddings input to the decoder. Except for the constant $d$ ($d = 128$ for most Transformers in NCO methods), the cross-attention between node embeddings $Z^{(l-1)}$ and $\widetilde{H}^{(l-1)}$ yields a complexity $\mathcal{O}(2\widetilde{n})$. Similarly, Eq.(4) exhibits the same linear complexity. By setting a fixed number of representative nodes, the cross-attention significantly reduces the computations between all nodes, which is more lightweight than prevailing self-attention-based Transformers in NCO methods.

## 4.3 SELF-IMPROVED TRAINING

Construction-based NCO models exhibit bias in decoding, where variations in starting nodes, destination nodes, and directions can result in vastly different solutions (Kwon et al., 2020). Benefiting from this bias, the model can gradually improve the solution quality by performing iterative local reconstruction until convergence (Luo et al., 2023; Ye et al., 2024). This paradigm holds significant potential to efficiently discover superior solutions without the need to explore complete solutions. However, current local reconstruction techniques still rely on SL or RL, which hinders their applicability to large-scale VRPs due to the scarcity of labels or the sparsity of rewards. Instead, we propose a Self-Improved Training (SIT) algorithm to specialize in local reconstruction for more effective solution exploration on large-scale VRPs. As illustrated in Figure 2, SIT involves iterative local reconstruction and model training, which are elaborated in the following.

**Local Reconstruction.** The local reconstruction comprises two steps. In the first step, a partial solution $\boldsymbol{\pi}^p$ of a random size $4 \leq \omega \leq l_{max}$ is sampled from $\boldsymbol{\pi}$, where $l_{max}$ refers to the maximum size of the partial solution. Since $\boldsymbol{\pi}$ can be expressed as a circle of nodes, we allow either clockwise or counterclockwise sampling direction. In the second step, the neural model reconstructs the partial solution node by node from its first node to the last node, i.e., the order of nodes between the first and the last nodes is rearranged. This generated partial solution $\boldsymbol{\pi}^{p\prime}$ is compared with $\boldsymbol{\pi}^p$, so that the better one (e.g., the one with shorter length) is adopted in the complete solution $\boldsymbol{\pi}$. In other words, a better partial solution can result in a better complete solution. Through iterative local reconstructions, the quality of the solution $\boldsymbol{\pi}$ can be significantly improved.

To improve the reconstruction efficiency, $M$ non-overlapping partial solutions are sampled and reconstructed in parallel. The overlap is avoided by evenly dividing the solution into $M$ consecutive segments of equal length $\omega$, in which the sampling and reconstruction can be parallelly performed. In this paper, we set $M$ to $\lfloor n/\omega \rfloor$ with $n$ denoting the problem size. Throughout the SIT process, we maintain a dataset $\mathcal{D}$ containing VRP instances and their solutions updated by local reconstructions. In each SIT iteration, the local reconstructions progressively enhance the quality of solutions in the dataset, which avoids revisiting previously explored solutions and thus improves the efficiency.

**Model Training.** The enhanced solutions from local reconstructions serve as pseudo-labels for training the model in a supervised manner. For a large-scale VRP instance (e.g., TSP instance with 100K nodes), the learning for constructing its complete solution can be difficult due to the massive GPU memory usage. To relieve the issue, we restrict the model's learning scope to local parts of the solution. Specifically, a random size $4 \leq \omega \leq l_{max}$ is adopted to sample partial solutions from the dataset $\mathcal{D}$. Let $\hat{\boldsymbol{\pi}}_{1:\omega}^p = (\hat{\pi}_1^p, \hat{\pi}_2^p, \ldots, \hat{\pi}_\omega^p)$ be a sampled partial solution. Taking it as a pseudo-label, the model learns to predict the order from $\hat{\pi}_1^p$ to $\hat{\pi}_\omega^p$ using the proposed Transformer network parameterized by $\boldsymbol{\theta}$. The loss function can be formulated as

$$\mathcal{L}(\boldsymbol{\theta}) = \mathbb{E}_{S^p \sim \mathcal{D}}[-\log p_{\boldsymbol{\theta}}(\hat{\pi}_t^p \mid S^p, \hat{\boldsymbol{\pi}}_{1:t-1}^p, \hat{\pi}_\omega^p)], \quad \forall t \in \{2, \ldots, \omega - 1\}, \tag{6}$$

where $\hat{\pi}_1^p$ and $\hat{\pi}_\omega^p$ denote the first and last nodes of the partial solution, and their embeddings constitute the representative node embeddings and graph node embeddings in Eq.(2). $\hat{\boldsymbol{\pi}}_{1:t-1}^p$ represents the sequence of visited nodes till the $t$-th decoding step. By training with the partial solutions, the model can be more efficient in enhancing solutions during the local reconstruction.

The SIT process alternates between the local reconstruction and the model training, until a predefined time budget is reached. The detailed training process and pseudocodes are provided in Appendix D.

## 5 EXPERIMENTS

We empirically evaluate the proposed method from two perspectives. Firstly, we compare the proposed method with diverse baseline methods to demonstrate its performance on synthetic and real-world large-scale TSP and CVRP instances. We then analyze the impact of the proposed method's critical components to verify our method's capability in reducing computational and memory complexities and analyze key hyperparameters.

**Dataset.** Following the common approach in literature Kool et al. (2019); Kwon et al. (2020); Luo et al. (2023), we generate five synthetic datasets with instances of scales 1K, 5K, 10K, 50K, and 100K, respectively. We denote TSP and CVRP instances of these scales as TSP/CVRP1K, 5K, 10K, 50K, and 100K, respectively. According to Fu et al. (2021), we set the number of instances for the TSP1K test dataset to 128. For datasets with larger instances, each contains 16 instances. Similarly, the CVRP test dataset includes the same number of instances, with capacities set to 250 for CVRP1K, 500 for CVRP5K, 1,000 for CVRP10K, and 2,000 for CVRP50K/100K. The optimal solutions of TSP instances are computed using LKH3 (Helsgaun, 2017), while CVRP instances are solved via HGS (Vidal, 2022). To evaluate our method on real-world large-scale instances, we also extract all symmetric instances with EUC_2D features and more than 1K nodes from TSPLIB Reinelt (1991) and CVRPLIB Uchoa et al. (2017), a total of 33 TSP instances and 14 CVRP instances.

**Model Setting&Training.** For our Transformer network, we set the embedding dimension $d = 128$. The decoder employs $L = 6$ stacked cross-attention modules, with each attention layer including 8 attention heads and a feed-forward layer with a hidden dimension of 512. The model initially undergoes a warm-up training process on instances of scale 1K using the pseudo-labels generated by random insertion (see Appendix D). After that, we continue the self-improved training on instances of scale 1K and then leverage the trained model to conduct separate training on larger scales, including 5K, 10K, 50K, and 100K.

The size $D$ of the training dataset for scales 1K, 5K/10K, 50K/100K are 20K, 200, and 100, respectively. Each iteration in our SIT algorithm comprises 100 times of local reconstruction and 20 epochs of model training. The Adam optimizer (Kingma & Ba, 2015) is utilized for training the models, with an initial learning rate 1e-4 and a decay rate 0.97 per epoch. Throughout the SIT process, the maximum length of partial solutions $l_{max}$ is 1,000 to balance efficiency and effectiveness. In all experiments, we use a single NVIDIA GeForce RTX 3090 GPU with 24GB memory for both training and testing. We provide more detailed training settings in Appendix E.

**Baselines.** We compare our method with **1) Classical Solvers:** Concorde (Applegate et al., 2006), LKH3 (Helsgaun, 2017), and HGS (Vidal, 2022); **2) Insertion Heuristic:** Random Insertion; **3) Construction-based NCO Methods:** POMO (Kwon et al., 2020), BQ (Drakulic et al., 2023), LEHD (Luo et al., 2023), INViT (Fang et al., 2024), and SIGD (Pirnay & Grimm, 2024); **4) Heatmap-based Methods:** DIFUSCO (Sun & Yang, 2023); **5) Decomposition-based Method:** GLOP (Ye et al., 2024), and H-TSP (Pan et al., 2023);**6) Local Policy-based Method:** ELG (Gao et al., 2024).

Table 1: Comparative results on synthetic TSP and CVRP instances. *: results are cited directly from original publications. N/A: the method exceeds the time limit (e.g., seven days) or produces infeasible solutions. OOM: the method exceeded memory limits.

| Method | TSP1K Obj. (Gap) | Time | TSP5K Obj. (Gap) | Time | TSP10K Obj. (Gap) | Time | TSP50K Obj. (Gap) | Time | TSP100K Obj. (Gap) | Time |
|---|---|---|---|---|---|---|---|---|---|---|
| LKH3 | 23.12 (0.00%) | 1.7m | 50.97 (0.00%) | 12m | 71.78 (0.00%) | 33m | 159.93 (0.00%) | 10h | 225.99 (0.00%) | 25h |
| Concorde | 23.12 (0.00%) | 1m | 50.95 (-0.05%) | 31m | 72.00 (0.15%) | 1.4h | N/A | N/A | N/A | N/A |
| Random Insertion | 26.11 (12.9%) | <1s | 58.06 (13.9%) | <1s | 81.82 (13.9%) | <1s | 182.65 (14.2%) | 15.4s | 258.13 (14.2%) | 1.7m |
| DIFUSCO* | 23.39 (1.17%) | 11.5s | – | – | 73.62 (2.58%) | 3.0m | – | – | – | – |
| H-TSP | 24.66 (6.66%) | 48s | 55.16 (8.21%) | 1.2m | 77.75 (8.38%) | 2.2m | OOM | | OOM | |
| GLOP | 23.78 (2.85%) | 10.2s | 53.15 (4.26%) | 1.0m | 75.04 (4.39%) | 1.9m | 168.09 (5.10%) | 1.5m | 237.61 (5.14%) | 3.9m |
| POMO aug×8 | 32.51 (40.6%) | 4.1s | 87.72 (72.1%) | 8.6s | OOM | | OOM | | OOM | |
| ELG aug×8 | 25.738 (11.33%) | 0.8s | 60.19 (18.08%) | 21s | OOM | | OOM | | OOM | |
| LEHD RRC1,000 | 23.29 (0.72%) | 3.3m | 54.43 (6.79%) | 8.6m | 80.90 (12.5%) | 18.6m | OOM | | OOM | |
| BQ bs16 | 23.43 (1.37%) | 13s | 58.27 (10.7%) | 24s | OOM | | OOM | | OOM | |
| SIGD bs16 | 23.36 (1.03%) | 17.3s | 55.77 (9.42%) | 30.5m | OOM | | OOM | | OOM | |
| INViT-3V greedy | 24.66 (6.66%) | 9.0s | 54.49 (6.90%) | 1.2m | 76.85 (7.07%) | 3.7m | 171.42 (7.18%) | 1.3h | 242.26 (7.20%) | 5.0h |
| LEHD greedy | 23.84 (3.11%) | 0.8s | 58.85 (15.46%) | 1.5m | 91.33 (27.24%) | 11.7m | OOM | | OOM | |
| BQ greedy | 23.65 (2.30%) | 0.9s | 58.27 (14.31%) | 22.5s | 89.73 (25.02%) | 1.0m | OOM | | OOM | |
| SIGD greedy | 23.573 (1.96%) | 1.2s | 57.19 (12.20%) | 1.8m | 93.80 (30.68%) | 15.5m | OOM | | OOM | |
| Ours greedy | 23.569 (1.95%) | 0.2s | 52.59 (3.17%) | 5.2s | 74.69 (4.05%) | 20.1s | 168.50 (5.36%) | 7.7s | 239.84 (6.13%) | 33.0m |
| Ours PRC10 | 23.396 (1.20%) | 0.9s | 52.36 (2.73%) | 5.1s | 73.99 (3.08%) | 10.0s | 166.69 (4.22%) | 1.33m | 235.38 (4.16%) | 3.0m |
| Ours PRC50 | 23.279 (0.69%) | 4.6s | 51.92 (1.85%) | 23.4s | 73.41 (2.27%) | 49.0s | 165.01 (3.17%) | 4.9m | 233.13 (3.16%) | 9.2m |
| Ours PRC100 | 23.254 (0.58%) | 9.4s | 51.82 (1.67%) | 52.0s | 73.29 (2.11%) | 1.7m | 164.59 (2.91%) | 8.6m | 232.55 (2.90%) | 17m |
| Ours PRC500 | 23.217 (0.43%) | 46s | 51.70 (1.43%) | 4.6m | 73.12 (1.87%) | 8.5m | 164.09 (2.60%) | 42.2m | 231.75 (2.55%) | 1.4h |
| Ours PRC1,000 | **23.207 (0.38%)** | 1.5m | **51.67 (1.36%)** | 9.4m | **73.08 (1.81%)** | 17.0m | **163.95 (2.51%)** | 1.38h | **231.52 (2.45%)** | 2.6h |

| Method | CVRP1K Obj. (Gap) | Time | CVRP5K Obj. (Gap) | Time | CVRP10K Obj. (Gap) | Time | CVRP50K Obj. (Gap) | Time | CVRP100K Obj. (Gap) | Time |
|---|---|---|---|---|---|---|---|---|---|---|
| HGS | 36.29 (0.00%) | 2.5m | 89.74 (0.00%) | 2.0h | 107.40 (0.00%) | 5.0h | 267.73 (0.00%) | 8.1h | 476.11 (0.00%) | 24h |
| LKH3 | 37.09 (2.21%) | 3.3m | 93.71 (5.19%) | 1.33h | 118.76 (10.6%) | 1.74h | 399.12 (49.1%) | 15.8h | N/A | N/A |
| Random Insertion | 57.42 (58.2%) | <1s | 154.38 (72.0%) | <1s | 191.80 (78.6%) | <1s | 490.56 (83.2%) | <1s | 943.87 (98.3%) | 2s |
| GLOP-G (LKH3) | 39.50 (8.83%) | 1.3s | 98.90 (10.2%) | 6.8s | 116.28 (8.27%) | 11.2s | OOM | | OOM | |
| POMO aug×8 | 84.89 (134%) | 4.8s | 393.27 (338%) | 11m | OOM | | OOM | | OOM | |
| ELG aug×8 | 41.57 (14.56%) | 1.1s | 109.54 (22.06%) | 30s | OOM | | OOM | | OOM | |
| LEHD RRC1,000 | 37.43 (3.15%) | 3.4m | 101.07 (12.6%) | 31m | 138.73 (29.2%) | 41m | OOM | | OOM | |
| BQ bs16 | 38.17 (5.17%) | 14s | 104.40 (16.3%) | 2.6m | OOM | | OOM | | OOM | |
| SIGD bs16 | 39.15 (7.91%) | 17.3s | 103.46 (15.3%) | 1.91m | 131.48 (22.4%) | 3.97m | 477.43 (78.3%) | 25.9m | OOM | |
| INViT-3V greedy | 42.75 (17.8%) | 11.4s | 109.85 (22.41%) | 1.4m | 141.41 (31.66%) | 4.2m | 402.05 (50.17%) | 2.9h | 688.80 (44.67%) | 8.3h |
| LEHD greedy | 38.91 (7.23%) | 0.8s | 105.61 (17.69%) | 1.56m | 146.24 (36.16%) | 11.85m | OOM | | OOM | |
| BQ greedy | 39.28 (8.23%) | 1.03s | 108.09 (20.48%) | 8.1s | 196.44 (82.9%) | 1.2m | OOM | | OOM | |
| SIGD greedy | 40.18 (10.7%) | 1.2s | 106.14 (18.3%) | 7.9s | 135.12 (25.8%) | 45s | 493.64 (84.4%) | 4.3m | OOM | |
| Ours greedy | 38.11 (5.01%) | 0.2s | 92.44 (3.01%) | 5.49s | 109.02 (1.50%) | 20.62s | 269.34 (0.60%) | 8.06m | 475.06 (-0.22%) | 33.1m |
| Ours PRC10 | 37.93 (4.52%) | 0.7s | 93.92 (4.65%) | 3.9s | 112.17 (4.43%) | 6.8s | 285.20 (6.52%) | 28s | 496.24 (4.23%) | 59s |
| Ours PRC50 | 37.57 (3.54%) | 3.5s | 92.06 (2.58%) | 19.9s | 108.79 (1.29%) | 34s | 271.77 (1.51%) | 2.3m | 476.71 (0.13%) | 4.8m |
| Ours PRC100 | 37.49 (3.31%) | 8.0s | 91.58 (2.05%) | 46s | 108.04 (0.59%) | 1.3m | 268.02 (0.11%) | 5.49m | 471.35 (-1.00%) | 11.5m |
| Ours PRC500 | 37.33 (2.88%) | 44.6s | 91.00 (1.41%) | 4.4m | 106.85 (-0.51%) | 7.6m | 263.56 (-1.56%) | 31.1m | 465.18 (-2.30%) | 1.1h |
| Ours PRC1,000 | **37.28 (2.72%)** | 1.5m | **90.81 (1.19%)** | 8.8m | **106.69 (-0.66%)** | 15.2m | **262.82 (-1.83%)** | 1.04h | **463.95 (-2.55%)** | 2.17h |

**Metrics&Inference.** For comparison, we provide the average objective value (Obj.), optimality gap (Gap), and inference time (Time) of each method. Obj. indicates the length of the VRP solution, with shorter values indicating better performance. Gap measures the solution difference from the ground truth (i.e., results produced by LKH for TSP and HGS for CVRP). Time, recorded in seconds (s), minutes (m), or hours (h), reflects the efficiency in generating solutions for test instances.

For our method, we present the results of the greedy search and Parallel local ReConstruction (PRC) under different numbers of iterations. For PRC, we adopt random insertion to generate initial solutions. We refer to Appendix F for the impact of different initialization methods on PRC performance.

## 5.1 COMPARATIVE RESULTS

We present the results on synthetic TSP and CVRP instances in Table 1. From the results, we can observe that our method consistently demonstrates superior performance. For both TSP and CVRP, when all baseline methods use the greedy search for inference, our method significantly outperforms the representative construction-based NCO methods with much smaller gaps and runtime. Our method, with only 50 PRC iterations, can beat the other learning-based methods in terms of both solution quality and solving efficiency across all scales except for CVRP1K. On CVRP1K, our method needs 500 or more PRC iterations to achieve the best among all the competitors. Overall, our method shows good scalability and can achieve outstanding performance even for very large-scale problem instances with up to 100K nodes. Remarkably, our method outperforms the classical solver HGS on CVRP10K, CVRP50K, and CVRP100K. As we know, this is the first time that a learning-based method gains a significant advantage over the specialized solver on large-scale VRP instances, manifesting a notable achievement for the NCO methods.

Table 2: Results on TSPLIB and CVRPLIB. OOM: The method is inapplicable due to the memory limit. [†]: There exist instances that are not solvable by the NCO method due to the OOM issue.

| | TSPLIB | | | | CVRPLIB | | | |
|---|---|---|---|---|---|---|---|---|
| Method | $1K < n \leq 5K$ | $n > 5K$ | All | Solved # | $1K < n \leq 7K$ | $n > 7K$ | All | Solved # |
| GLOP | 5.017% | 6.870%[†] | 5.495% | 31/33 | 15.335% | 21.317% | 17.898% | 14/14 |
| ELG aug×8 | 11.34% | OOM | 11.34% | 23/33 | 10.57%[†] | OOM | 10.57% | 6/14 |
| BQ bs16 | 10.648% | 30.579%[†] | 12.948% | 26/33 | 13.918% | OOM | 13.918% | 8/14 |
| LEHD RRC1,000 | 3.996% | 18.458%[†] | 7.371% | 30/33 | 8.423% | 21.525%[†] | 11.043% | 10/14 |
| SIGD greedy | 12.369% | 152.879%[†] | 48.630% | 31/33 | 14.733% | 49.491% | 29.629% | 14/14 |
| BQ greedy | 11.640% | 162.116%[†] | 64.649% | 32/33 | 16.923% | 52.267% | 32.071% | 14/14 |
| INViT greedy | 11.492% | 9.996% | 11.038% | 33/33 | 15.873% | 26.637% | 20.486% | 14/14 |
| LEHD greedy | 11.139% | 39.343%[†] | 17.720% | 30/33 | 15.203% | 32.797%[†] | 18.722% | 10/14 |
| Ours greedy | 6.767% | 10.697% | 8.244% | 33/33 | 15.806% | 15.504% | 15.677% | 14/14 |
| Ours PRC1,000 | **1.576%** | **4.043%** | **2.556%** | 33/33 | **8.347%** | **11.209%** | **9.574%** | 14/14 |

Table 3: Effects of self-improved training and cross-attention. w/o SIT: Our model is trained by SL rather than SIT. w/o Cross-Attention: Our model uses the conventional self-attention rather than cross-attention. Ours: The model is equipped with both cross-attention and SIT. The trained models are tested by using greedy search. Gap (%), Time (s), and Memory (MB) are averaged over instances.

| Scale | 1K | | | 5K | | | 10K | | |
|---|---|---|---|---|---|---|---|---|---|
| | Gap | Time | Memory | Gap | Time | Memory | Gap | Time | Memory |
| w/o SIT | 12.69 | 0.21 | 9.65 | 24.98 | 5.41 | 47.94 | 37.63 | 19.83 | 94.41 |
| w/o Cross-Attention | 5.20 | 0.75 | 96.94 | 6.04 | 82.22 | 2317.50 | 12.03 | 705.90 | 9219.09 |
| Ours | **5.01** | **0.21** | **9.65** | **5.37** | **5.41** | **47.94** | **7.58** | **19.83** | **94.41** |

The experimental results on real-world large-scale TSPLIB and CVRPLIB instances are provided in Table 2. When all methods perform the greedy search for inference, our method achieves smaller gaps than the other construction-based NCO methods on 2 (out of 4) groups of instances. Using the PRC method, our method consistently outperforms the other baseline methods, achieving the smallest gaps for both TSP and CVRP. While some NCO methods struggle to solve large-scale instances due to heavy self-attention-based Transformers, our method can solve all TSP and CVRP instances with up to $85,900$ nodes.

## 5.2 ABLATION STUDY

We conduct an ablation study to demonstrate the impact of the key components (i.e., cross-attention and SIT) of our method. First, we train the proposed cross-attention-based Transformer on large-scale VRPs without the SIT algorithm (i.e., w/o SIT in Table 3). Due to the scarce labeled data on larger VRP instances, we train the model on CVRP100 following (Luo et al., 2023). Then, we replace the cross-attention in our Transformer network with the self-attention, and the resultant model (i.e., w/o Cross-Attention in Table 3) is trained by our SIT algorithm. We test the above two models on CVRP1K/5K/10K. From Table 3, we observe that our model trained with SIT significantly outperforms the model trained without SIT, which indicates SIT enables more effective model training on large-scale problems. On the other hand, the results verify that the proposed cross-attention is more efficient than the self-attention in terms of both time and memory usage (e.g., over $30\times$ speedup and $\approx 100\times$ memory saving on CVRP10K).

## 5.3 ADDITIONAL ANALYSIS

**Efficiency analysis.** We test our method and self-attention-based Transformers, including POMO, LEHD, SIGD Kwon et al. (2020); Luo et al. (2023); Pirnay & Grimm (2024), on TSP instances of different scales, and compare them in terms of inference time and memory usage. The results in Table 4 demonstrate the superior time and space efficiency of our Transformer network across all problem scales. For example, our model runs $4\times$ faster and takes up $10\times$ smaller memory than LEHD on instances with 1K nodes. Moreover, the time and space efficiency significantly enlarge as the problem scale grows. Especially for instances with 50K or 100K nodes, self-attention-based

Table 4: The time and memory usage of our model and representative NCO methods. All methods are tested by using greedy search. Time (s) and Memory (MB) are averaged over instances.

| Scale | 1K | | 5K | | 10K | | 50K | | 100K | |
|---|---|---|---|---|---|---|---|---|---|---|
| | Time | Memory | Time | Memory | Time | Memory | Time | Memory | Time | Memory |
| POMO | 0.5 | 107.5 | 192.6 | 2599.1 | 553.2 | 10352.3 | OOM | | OOM | |
| SIGD | 1.2 | 41.2 | 108.0 | 971.0 | 930.6 | 3846.8 | OOM | | OOM | |
| LEHD | 0.8 | 97.4 | 90.0 | 2323.2 | 730.8 | 9224.3 | OOM | | OOM | |
| Ours | **0.2** | **9.3** | **5.5** | **45.4** | **20.6** | **91.9** | **483.4** | **459.4** | **1983.3** | **918.6** |

Table 5: Impact of $l_{max}$ in both training and inference.

| $l_{max}$ (Inference) $\rightarrow$ | 100 | 1,000 | 10,000 |
|---|---|---|---|
| $l_{max}$ (Training) $\downarrow$ | Gap (Time) | Gap (Time) | Gap (Time) |
| 100 | 4.568% (18s) | 2.647% (1.7m) | 2.731% (6.0m) |
| 1,000 | 4.513% (18s) | 2.319% (1.7m) | 2.279% (6.0m) |
| 10,000 | 4.566% (18s) | 2.490% (1.7m) | 2.365% (6.0m) |

models encounter out-of-memory (OOM) issues, highlighting their limitations in tackling large-scale problems. In contrast, our model is consistently efficient on large-scale VRPs, indicating its advantageous scalability. Furthermore, the time and memory usage in the methods empirically verify the linear complexity of our cross-attention-based Transformer and the quadratic complexities of self-attention-based models.

**Sensitivity analysis.** We evaluate the impact of the key hyperparameter $l_{max}$ on model performance. To this end, we train and test (with PRC100) our model with different values of $l_{max}$ on TSP10K. As shown in Table 5, the model trained to construct solutions with $l_{max} = 10,000$ delivers inferior policy compared to the one trained with $l_{max} = 1,000$. Intuitively, learning to construct overly long solutions overwhelms the model's capability, which makes it hard to learn favorable representations by a limited number of cross-attention computations. On the other hand, setting $l_{max}$ to small values leads to excessively local policies and sacrifices global performance. Therefore, we set $l_{max}$ to an intermediate value of 1,000 for balancing the training and inference performance.

## 6 CONCLUSION

In this paper, we have proposed a lightweight cross-attention mechanism with linear complexity to improve the efficiency of the NCO model in solving large-scale VRPs. Benefiting from propagating node embeddings through representative nodes, the cross-attention mechanism maintains effective interactions between nodes while achieving low complexity. In addition, we have developed a novel Transformer network to learn efficient and favorable solutions, in which the cross-attention is iteratively used to advance the node embeddings. Moreover, we have proposed an innovative Self-Improved Training (SIT) algorithm for direct model training on large-scale VRP instances without the need for labeled data. Extensive experimental results on TSP and CVRP with up to 100K nodes in both synthetic and real-world distributions fully demonstrate the superior performance of our method. Since a specific range of random sizes is predefined when sampling the partial solutions, a potential improvement is to develop an adaptive strategy for setting the sampling size of partial solutions, thereby enabling more efficient model training and PRC. In addition, we will extend our method to other types of combinatorial optimization problems, such as scheduling, bin packing, and knapsack problems.

## ACKNOWLEDGMENTS

This work was supported by the National Natural Science Foundation of China (Grant No. 62476118), the Natural Science Foundation of Guangdong Province (Grant No. 2024A1515011759), the National Natural Science Foundation of Shenzhen (Grant No.JCYJ20220530113013031), the Research Grants Council of the Hong Kong Special Administrative Region, China (CityU11215622 and CityU11215723).

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

# A    COMPLEXITY ANALYSIS OF ITERATIVE RECONSTRUCTION

We would like to clarify that the complexity analysis in the main text (Section 4.2) is to demonstrate the complexity advantage of our model rather than the entire algorithm. Our model has linear complexity (i.e., $O(n)$) and is more suitable for handling large-scale problems than existing NCO models, with significant computational efficiency compared to the quadratic complexity (i.e., $O(n^2)$) of self-attention-based models in the literature.

For the iterative reconstruction process, the computational complexity is $\mathcal{O}(kl_{max}^2)$ and space complexity is $\mathcal{O}(n)$, where $k$ is the number of iterations and $l_{max}$ is the maximum length of the partial solutions during reconstruction progress. Specifically, a single reconstruction has $\mathcal{O}(l_{max}^2)$ computational complexity, since 1) it consists of $l_{max}$ sequential model predictions and 2) one time of linear model prediction has $\mathcal{O}(l_{max})$ complexity. Therefore, performing $k$ reconstructions yields an overall computational complexity of $\mathcal{O}(kl_{max}^2)$. As the space complexity is not cumulative, the space required for performing $k$ reconstructions remains $\mathcal{O}(l_{max})$. In this paper, $\frac{n}{l_{max}}$ reconstructions are executed parallelly in each iteration. Therefore, the space complexity is $\mathcal{O}(\frac{n}{l_{max}} \cdot l_{max}) = \mathcal{O}(n)$, while the overall computational complexity remains $\mathcal{O}(kl_{max}^2)$.

It is worth noting that many NCO methods (including ours) employ post-search or iterative inference to trade more computation time for better accuracy. To comprehensively illustrate our method's advantages in computational efficiency, we compare it with three representative NCO methods (i.e., POMO (Kwon et al., 2020), LEHD (Luo et al., 2023), and SIGD (Pirnay & Grimm, 2024)) under single-round (greedy search) and multi-round inference, respectively. Under the multi-round inference, we use augmentation (aug) for POMO, random reconstruction (RRC) for LEHD, and beam search (bs) for SIGD as suggested in their original papers, and each of them is operated with 20 rounds.

Table 6: Complexity comparison between our method and three representative ones with single-round inference

| Method | TSP1K | | | TSP5K | | | TSP10K | | |
|---|---|---|---|---|---|---|---|---|---|
| | Gap | Time | Memory | Gap | Time | Memory | Gap | Time | Memory |
| POMO greedy | 42.3% | 0.5s | 108MB | 72.9% | 3.2m | 2599MB | 86.48% | 9.2m | 10352MB |
| SIGD greedy | 1.96% | 1.2s | 41MB | 12.20% | 1.8m | 971MB | 30.68% | 15.5m | 3847MB |
| LEHD greedy | 3.11% | 0.8s | 97MB | 15.46% | 1.5m | 2323MB | 27.24% | 11.7m | 9224MB |
| Ours greedy | **1.95%** | **0.2s** | **9.3MB** | **3.17%** | **5.2s** | **45MB** | **4.05%** | **20s** | **92MB** |

Table 7: Complexity comparison between our method and three representative ones with multi-round inference

| Method | TSP1K | | | TSP5K | | | TSP10K | | |
|---|---|---|---|---|---|---|---|---|---|
| | Gap | Time | Memory | Gap | Time | Memory | Gap | Time | Memory |
| POMO aug×20 | 40.4% | 10.6s | 880MB | 71.8% | 21.4m | 21185MB | | OOM | |
| SIGD bs20 | 1.01% | 21.7s | 750MB | 9.15% | 36.8m | 17564MB | | OOM | |
| LEHD RRC20 | 1.70% | 3.7s | 97MB | 12.09% | 1.6m | 2323MB | 21.78% | 12.1m | 9224MB |
| Ours PRC20 | **0.92%** | **1.6s** | **12MB** | **2.27%** | **8.6s** | **85MB** | **2.59%** | **20.3s** | **129MB** |

The experimental results with the single-round inference are given in Table 6. These results indicate that our method achieves better optimality gaps with less running time and memory usage on all three problem sizes. Our method yields at least $2.5\times$, $17.3\times$, and $27.6\times$ speedups and $4.4\times$, $21.5\times$, and $41.8\times$ memory reductions on TSP1K, TSP 5K, and TSP10K, respectively.

Table 7 provides the experimental results under the multi-round inference setting. These results show that our method also significantly outperforms the other three competitors regarding the optimality gap, running time, and memory usage. Our method achieves at least $2.3\times$, $11.1\times$, and $35.7\times$ speedups and $8\times$, $27.3\times$, and $71.5\times$ memory reductions on TSP1K, TSP5K, and TSP10K, respectively.

In short, the advantage of computational complexity enables our method to have higher solving efficiency in either a greedy or iterative manner, especially when dealing with problem instances with sizes larger than 1K.

# B  IMPLEMENTATION DETAILS FOR CVRP

In this section, we introduce the problem setup, Transformer network details, complexity analysis, and PRC implementation details for CVRP.

## B.1  PROBLEM SETUP

A CVRP instance comprises one depot node and $n$ customer nodes, where each customer node $i$ has a demand $\delta_i$ to fulfill. Our goal is to find a set of sub-tours that begin and end at the depot, ensuring the sum of demands in each sub-tour adheres to the vehicle's capacity constraint $C$. The objective is to minimize the total distance across these sub-tours while maintaining the capacity constraint $C$. Our CVRP instances are generated in a manner similar to that described in (Kool et al., 2019), featuring customer and depot node coordinates uniformly sampled from a unit square $[0, 1]^2$. Demands $\delta_i$ are uniformly sampled from $1, \ldots, 9$.

In line with (Kool et al., 2022; Drakulic et al., 2023; Luo et al., 2023), we establish a feasible CVRP solution formation. Instead of isolating a depot visit as a distinct step, we employ binary variables to signify whether a customer node is accessed via the depot or another customer node. In a feasible solution, a node is assigned 1 if accessed through the depot and 0 if accessed through another customer node. For example, a viable CVRP solution $\{0, 1, 2, 3, 0, 4, 5, 0, 6, 7, 0, 8, 9, 10\}$ with 0 representing the depot can be represented as shown below:

$$\begin{bmatrix} 1 & 2 & 3 & 4 & 5 & 6 & 7 & 8 & 9 & 10 \\ 1 & 0 & 0 & 1 & 0 & 1 & 0 & 1 & 0 & 0 \end{bmatrix}, \tag{7}$$

where the first row displays the visited node sequence, while the second row signifies if each node is accessed through the depot or another customer node.

This notation aims to maintain solution consistency. In CVRP cases, solutions with equal customer node counts might have differing sub-tour quantities, causing potential misalignment. This notation prevents such problems.

## B.2  TRANSFORMER NETWORK

**Embedding Layer.** In CVRP, the node feature $\mathbf{x}_i$ is a 3D vector, combining 2D coordinates and the demand of node $i$, where $\mathbf{x}_0$ and $\{\mathbf{x}_i\}_{i=1}^n$ are depot node and customer node features, respectively. The depot's demand is set as 0. We normalize the vehicle capacity $C$ to $\hat{C} = 1$ and the demand $\delta_i$ to $\hat{\delta}_i = \frac{\delta_i}{C}$ for simplicity. Given the node features $\{\mathbf{x}_i\}_{i=0}^n$, the encoder produces each node's embedding $\{\mathbf{h}_i\}_{i=0}^n$ by a linear projection such that $\mathbf{h}_i = \mathbf{x}_i W^{(0)} + \mathbf{b}^{(0)}, \forall i \in \{0, \ldots, n\}$, where $W^{(0)} \in \mathbb{R}^{3 \times d}$ and $\mathbf{b}^{(0)} \in \mathbb{R}^d$ are learnable matrices.

**Decoder with Cross-Attention Modules.** Similar to Kwon et al. (2020), we add the dynamically changing remaining capacity to the first and last node embeddings in the decoder. The remaining capacity can be denoted as $C_r \in \mathbb{R}^1$, and the first and last node embeddings are $\mathbf{h}_{\pi_1}$ and $\mathbf{h}_{\pi_{t-1}}$, respectively. We fuse the remaining capacity to the first node and last nodes' embeddings via

$$\begin{aligned} \mathbf{h}'_{\pi_1} &= [\mathbf{h}_{\pi_1}, C_r]W_1 + \mathbf{b}_1 \\ \mathbf{h}'_{\pi_{t-1}} &= [\mathbf{h}_{\pi_{t-1}}, C_r]W_2 + \mathbf{b}_2, \end{aligned} \tag{8}$$

where $W_1, W_2 \in \mathbb{R}^{(d+1) \times d}$ and $\mathbf{b}_1, \mathbf{b}_2 \in \mathbb{R}^d$. Then the representative nodes' initial embeddings $Z^{(0)}$ and graph node embeddings $\widetilde{H}^{(0)}$ are calculated as

$$Z^{(0)} = [\mathbf{h}'_{\pi_1}, \mathbf{h}'_{\pi_{t-1}}], \quad \widetilde{H}^{(0)} = [\mathbf{h}'_{\pi_1}, \mathbf{h}'_{\pi_{t-1}}, H_a^t], \tag{9}$$

where $W_1, W_2 \in \mathbb{R}^{d \times d}$ are learnable matrices, $H_a^t$ is the set of unvisited customer nodes' embeddings at the $t$-th step. In the first decoding step, the depot node $\pi_1$ is selected to be the initial partial solution, and we treat it as both the first and last nodes for the second decoding step.

Similar to TSP, the output of our model's $L$-th linear attention module is $Z^{(L)}$ and $\widetilde{H}^{(L)}$, where $\widetilde{H}^{(L)} = \{\widetilde{\mathbf{h}}_i^{(L)}\}$. Then, a linear projection and softmax function are applied to it, producing the selected probability of each unvisited node. The first and last nodes are masked before the softmax calculation, i.e.,

$$\mathbf{u}_i = \begin{cases} \widetilde{\mathbf{h}}_i^{(L)} W_O, & i \neq 1 \text{ or } 2 \\ -\infty, & \text{otherwise} \end{cases}, \tag{10}$$

where $W_O \in \mathbb{R}^{d \times 2}$ is a learnable matrix. Each $\mathbf{u}_i \in \mathbb{R}^2$ corresponds to two actions: access customer node $i$ via the depot or another customer node. It corresponds to the notation in Eq.(7). Finally, a softmax function is applied to all the $\mathbf{u}_i$ to produce the selected probability of each action.

**Complexity Analysis.** According to Eq.(3), the dimensions of input to cross-attention modules are $\mathbb{R}^{2 \times d}$ for $Z^{(l-1)}$ and $\mathbb{R}^{\widetilde{n} \times d}$ for $\widetilde{H}^{(l-1)}$, where $\widetilde{n} \leq (n+2)$ is the number of node embeddings input to the decoder. Except for the constant $d$ ($d = 128$ for most Transformers of NCO methods), the cross-attention between node embeddings $Z^{(l-1)}$ and $\widetilde{H}^{(l-1)}$ yields a complexity $\mathcal{O}(2\widetilde{n})$. Therefore, Eq.(4) exhibits the same linear complexity in solving CVRP.

### B.3 IMPLEMENTATION DETAILS OF PRC FOR SOLVING CVRP

During the PRC process, we need to ensure that the sampled partial solutions have no overlap, and that no illegal solutions are generated after merging the partial solutions. These details can be described below.

**Guarantee Non-overlapping.** We always sample a contiguous subset of the complete solution as a partial solution. All partial solutions have the same number of customer nodes. We sample these partial solutions sequentially from the beginning of the solution to the end, following its order. Each sampled partial solution is unique.

For example, consider a CVRP instance with ten customer nodes, represented by the solution $(0, 1, 2, 3, 4, 5, 0, 6, 7, 0, 8, 9, 10, 0)$, where $0$ represents the depot. If each partial solution contains four customer nodes, then the partial solutions can be $(0, 1, 2, 3, 4)$ and $(5, 0, 6, 7, 0, 8)$. The remaining segment, $(9, 10, 0)$, with only two customer nodes, will not participate in the reconstruction process.

**Avoid Violating the Capacity Constraint.** For CVRP, the sum of customer demands in each sub-tour must not exceed the vehicle's capacity. This constraint may make the solution infeasible after merging reconstructed partial solutions. To ensure the solutions remain feasible, we take two measures during the reconstruction:

**(a) Exclude Tail Subtour.** If the subtour at the end of a partial solution does not conclude at the depot, it will not participate in the reconstruction process.

For example, consider a CVRP solution $(0, 1, 2, 3, 4, 5, 0, 6, 7, 0, 8, 9, 10, 0)$, for its partial solution $(5, 0, 6, 7, 0, 8)$, the subtour $(0, 8)$ does not conclude at the depot, and it will not participate in the reconstruction.

If the tail subtour $(0, 8)$ participates in the reconstruction process, the result might be $(5, 0, 6, 7, 8)$, which is then merged with $(9, 10, 0)$ to form the partial solution $(5, 0, 6, 7, 8, 9, 10, 0)$. However, within it, $(0, 6, 7, 8, 9, 10, 0)$ might violate the capacity constraint. This occurs when the sum of the demands of node $\{8, 9, 10\}$ is very close to or equal to the capacity. Therefore, we need to exclude the reconstruction of the tail subtour.

**(b) Initial Capacity Calculation.** When the head node of the partial solution is not the depot, the vehicle's initial capacity is determined by the full capacity minus the total demand of nodes between this head node and its preceding depot in the original complete solution.

For example, consider the CVRP solution $(0, 1, 2, 3, 4, 5, 0, 6, 7, 0, 8, 9, 10, 0)$ again, for the partial solution $(5, 0, 6, 7, 0, 8)$, its head node is not the depot. The nodes between this head node and the preceding depot are $\{1, 2, 3, 4\}$. Therefore, the initial capacity for reconstructing the partial solution $(5, 0, 6, 7, 0, 8)$ is the full capacity minus the total demand of nodes $\{1, 2, 3, 4\}$.

If this constraint restriction is removed (i.e., using full capacity), the reconstructed partial solution $(5, 0, 6, 7, 0, 8)$ may become $(5, 6, 7, 8)$. When it merges with $(0, 1, 2, 3, 4)$, forming the route $(0, 1, 2, 3, 4, 5, 6, 7, 8)$, it has a high probability of exceeding the vehicle's capacity limit. To avoid this, we need to implement this initial capacity calculation.

## C    CHOICE OF REPRESENTATIVE NODES

In this paper, we use the first and last visited nodes as the representative node. We also try other options for the representative nodes, such as the pooling nodes (i.e., summing up the embeddings of all nodes and then averaging them) and learnable nodes (i.e., learning two embeddings). However, we experimentally find that models with these types of representative nodes suffer from numerical instability during training, where the loss becomes NaN in the early stage of training.

In solving VRPs, the last node is dynamically updated to the one selected in the previous step, and the number of unvisited nodes decreases. This adjustment continuously alters the relationship among the first, last, and unvisited nodes. The representative nodes must accurately reflect these changes at every step. However, pooling nodes or learnable nodes cannot achieve it. Instead, the first and last nodes inherently represent dynamically changing information and, thus, are more suitable as representative nodes.

Inspired by Deudon et al. (2018), we explore strategies to enhance the model's perception of the last node's information to improve the model's learning capabilities. We repeat the last node several times in the representative nodes, and conduct the model training on TSP1K instances with two SIT iterations. The optimality gap of corresponding models versus the number of repeated last nodes is plotted in Figure 3. We can find that there is a positive correlation between the number of repetitions and performance gains. Moreover, as the number of repetitions increases, the incremental benefits become smaller and smaller, especially when the number is larger than 15. In addition, increasing the number of representative nodes results in extra computational overhead. Considering these factors, we use 15 in the experiments to balance between performance improvement and resource efficiency.

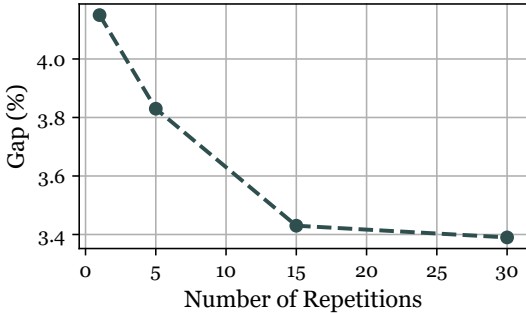

Figure 3: Comparion results with different numbers of repetitions on the last node.

## D PSEUDOCODE

This section provides the pseudocode-based explanation of SIT in Section 4.3 to enhance clarity and understanding. Algorithm 1 is the pseudocode-based description of SIT. Algorithm 2 is for generating the initial pseudo-labels. Algorithm 3 is for the parallel local reconstruction for a single instance. Algorithm 4 is for the model training process.

---

**Algorithm 1** Self-Improved Training

---

1: **Input:** Problem size $N$, dataset size $D$, batch size $B$, maximum length of partial solution $l_{max}$.
2: **Output:** The trained model with parameters $\boldsymbol{\theta}^*$.
3: Randomly initialize the model with parameters $\boldsymbol{\theta}$;
4: $S_i \sim \text{SampleInstance}(N)$    $\forall i \in \{1, \ldots, D\}$;
5: $\boldsymbol{\pi}_i \leftarrow \text{GenerateInitialSolution}(S_i)$    $\forall i \in \{1, \ldots, D\}$;          ▷ Serve as initial pseudo-labels.
6: **for** each iteration **do**
7:     $\boldsymbol{\pi}_i \leftarrow \text{LocalReconstruction}(\boldsymbol{\theta}, \boldsymbol{\pi}_i, S_i, N, l_{max})$    $\forall i \in \{1, \ldots, D\}$;
8:     $\boldsymbol{\theta} \leftarrow \text{ModelTraining}(\boldsymbol{\theta}, \{\boldsymbol{\pi}_i\}, \{S_i\}, D, B, l_{max})$    $\forall i \in \{1, \ldots, D\}$;
9: **end for**

---

---

**Algorithm 2** GenerateInitialSolution

---

1: **Input:** Instance $S$.                ▷ The procedure is essentially conducting random insertion.
2: **Output:** Complete solution $\boldsymbol{\pi}$.
3: Calculate the distance matrix $[d_{ij}]$;
4: Pick a random node to form the initial partial solution $\boldsymbol{\pi}$;
5: **while** there are unvisited nodes **do**
6:     Select a random unvisited node $i$
7:     Find the best position $(j, k)$ to insert $i$ that minimizes $d_{ji} + d_{ik} - d_{jk}$
8:     $\boldsymbol{\pi} \leftarrow$ Insert $i$ between nodes $j$ and $k$ in $\boldsymbol{\pi}$
9: **end while**

---

---

**Algorithm 3** LocalReconstruction

---

1: **Input:** Model parameter $\boldsymbol{\theta}$, solution $\boldsymbol{\pi}$, instance $S$, problem size $N$, maximum length of the partial solution $l_{max}$.
2: **Output:** The improved solution $\boldsymbol{\pi}$.
3: **for** each iteration **do**
4:     $\omega \sim \text{Uniform}[4, l_{max}]$;
5:     $\{\boldsymbol{\pi}_i^p, S_i^p\} \leftarrow \text{SamplePartialSolutions}(\boldsymbol{\pi}, S, \omega)$    $\forall i \in \{1, \ldots, \lfloor N/\omega \rfloor\}$;
6:     $\{\boldsymbol{\pi}_i^{p\prime}\} \leftarrow \text{ReconstructByModel}(\boldsymbol{\theta}, \{\boldsymbol{\pi}_i^p, S_i^p\})$    $\forall i \in \{1, \ldots, \lfloor N/\omega \rfloor\}$;
7:     $\boldsymbol{\pi}_i^p \leftarrow \text{SaveBetterOne}(\boldsymbol{\pi}_i^p, \boldsymbol{\pi}_i^{p\prime})$    $\forall i \in \{1, \ldots, \lfloor N/\omega \rfloor\}$;
8:     $\boldsymbol{\pi} \leftarrow \text{Combine}(\{\boldsymbol{\pi}_i^p\})$;
9: **end for**

---

---

**Algorithm 4** ModelTraining

---

1: **Input:** Model parameter $\boldsymbol{\theta}$, pseudo-labels $\{\boldsymbol{\pi}_i\}$, instances $\{S_i\}$, dataset size $D$, training batch size $B$, maximum length of the partial solution $l_{max}$.
2: **Output:** The trained model parameter $\boldsymbol{\theta}$.
3: **for** each epoch **do**
4:     **for** $step = 1, \ldots, \lfloor D/B \rfloor$ **do**
5:         $\omega \sim \text{Uniform}[4, l_{max}]$;
6:         $\hat{\boldsymbol{\pi}}_i^p, S_i^p \leftarrow \text{SamplePartialSolution}(\boldsymbol{\pi}_i, S_i, \omega) \quad \forall i \in \{1, \ldots, B\}$;
7:         **for** $t = 2, \ldots, \omega - 1$ **do**
8:             $\nabla \mathcal{L}(\boldsymbol{\theta}) \leftarrow \frac{1}{B} \sum_{i=1}^{B} \left[ -\nabla_{\boldsymbol{\theta}} \log p_{\boldsymbol{\theta}} \left( \hat{\pi}_{i_t}^p \mid S_i^p, \hat{\boldsymbol{\pi}}_{i_{1:t-1}}^p, \hat{\pi}_{i_\omega}^p \right) \right]$;
9:             $\boldsymbol{\theta} \leftarrow \text{ADAM}(\boldsymbol{\theta}, \nabla \mathcal{L}(\boldsymbol{\theta}))$;
10:         **end for**
11:     **end for**
12: **end for**

---

## E   TRAINING SETTING

Our Transformer networks are trained on instances with different scales separately. We summarize the training settings in Table 8:

- **Dataset Size.** It refers to the number of instances in the dataset, which varies depending on the problem scale. When training on a larger scale (e.g. 100K), fewer instances are required because a complete solution of this scale can generate enough partial solutions to facilitate efficient training.

- **Self-Improved Iteration.** Each self-improved training iteration involves a number of training epochs and a number of reconstruction iterations.

- **Training Epoch.** When all training instances have been used for training the model once, one epoch is completed.

- **Training Batch Size.** It refers to the number of training data used each time the model parameters are updated. In one training epoch, the dataset is divided into multiple smaller batches, each containing a certain number of instances. During training, larger-scale datasets like TSP100K and CVRP100K use a smaller batch size (i.e., 16) due to device memory constraints, while smaller-scale datasets like TSP1K and CVRP1K use a larger batch size (i.e., 256).

- **Reconstruction Iteration.** One iteration refers to the process in which all training instances are reconstructed once.

- **Reconstruction Batch Size.** In one local reconstruction iteration, the entire dataset is reconstructed in batches. Larger-scale datasets like TSP100K and CVRP100K use a smaller batch size due to device memory constraints, while smaller-scale datasets like TSP1K and CVRP1K use a larger batch size.

- **Learning Rate.** The initial learning rate for training the model.

- **Decay per Training Epoch.** The rate at which the learning rate decays with each epoch.

- **Maximum Length of Partial Solution.** This represents the maximum length for partial solutions during training and reconstruction.

In each self-improved iteration, we save the model that with the best greedy performance on the validation dataset to use in the next training iteration.

Table 8: Training settings.

|  | TSP1K | TSP5K | TSP10K | TSP50K | TSP100K |
|---|---|---|---|---|---|
| Dataset Size | 20,000 | 200 | 200 | 100 | 100 |
| Self-improved Iteration | 8 | 8 | 3 | 3 | 7 |
| Training Epoch | 20 | 20 | 20 | 20 | 20 |
| Training Batch Size | 256 | 32 | 32 | 32 | 16 |
| Reconstruction Iteration | 100 | 100 | 100 | 100 | 100 |
| Reconstruction Batch Size | 256 | 32 | 32 | 16 | 8 |
| Learning Rate | 1e-4 | 1e-4 | 1e-4 | 1e-4 | 1e-4 |
| Decay per Training Epoch | 0.97 | 0.97 | 0.97 | 0.97 | 0.97 |
| Partial Solution's Maximum Length | 1,000 | 1,000 | 1,000 | 1,000 | 1,000 |
|  | CVRP1K | CVRP5K | CVRP10K | CVRP50K | CVRP100K |
| Dataset size | 20,000 | 200 | 200 | 100 | 100 |
| Self-improved Iteration | 8 | 8 | 4 | 4 | 7 |
| Training Epoch | 20 | 20 | 20 | 20 | 20 |
| Training Batch Size | 256 | 32 | 32 | 32 | 16 |
| Reconstruction Iteration | 100 | 100 | 100 | 100 | 100 |
| Reconstruction Batch Size | 256 | 32 | 32 | 16 | 8 |
| Learning Rate | 1e-4 | 1e-4 | 1e-4 | 1e-4 | 1e-4 |
| Decay per Training Epoch | 0.97 | 0.97 | 0.97 | 0.97 | 0.97 |
| Partial Solution's Maximum Length | 1,000 | 1,000 | 1,000 | 1,000 | 1,000 |

## F Impact of Solution Initialization Methods on PRC performance

We empirically study the impact of different initialization methods on the PRC performance. We test three simplest initialization methods, including greedy search performed by the model, heuristic nearest neighbor, and heuristic random insertion. We refer to (Kool et al., 2019) for their details. We use PRC100 to improve the solutions generated by these initialization methods and obtain the results, which are shown in table 9. These results demonstrate that random insertion+PRC produces solutions with lower objective values than the other methods on 6 (out of 10) groups of instances. At the same time, random insertion is generally faster than the model's greedy search in initializing the solution (see Table 1). Considering these factors, we choose the random insertion method to generate the initial solution for PRC.

Table 9: Performance of PRC100 (in terms of Obj.) with different solution initialization methods.

|  | TSP1K | TSP5K | TSP10K | TSP50K | TSP100K |
|---|---|---|---|---|---|
| Model's greedy | **23.251** | **51.698** | 73.368 | 165.407 | 234.651 |
| Nearest neighbor | 23.254 | 54.954 | 79.159 | 179.886 | 255.367 |
| Random insertion | 23.254 | 51.824 | **73.293** | **164.591** | **232.550** |
|  | CVRP1K | CVRP5K | CVRP10K | CVRP50K | CVRP100K |
| Model's greedy | **37.460** | 91.857 | 108.229 | 268.416 | **464.325** |
| Nearest neighbor | 37.493 | 91.725 | 111.832 | 316.338 | 529.006 |
| Random insertion | 37.491 | **91.580** | **108.038** | **268.024** | 471.348 |

## G Comparison with TAM

We compare the proposed method with the decomposition-based method TAM Hou et al. (2023). Since TAM does not release source code, we test our method on the test dataset introduced in their paper. TAM's results are from the original paper. For our method, the model is trained on CVRP1K instances and subsequently tested across larger-scale instances. The results are shown in Table 10. We can observe that even with the simple greedy search, our method can achieve better objective values than these methods, indicating excellent performance and efficiency of our method in solving large-scale CVRP instances.

Table 10: Experimental results compared with the decomposition-based methods.

| Method | CVRP1K
Obj. (Time) | CVRP2K
Obj. (Time) | CVRP5K
Obj. (Time) | CVRP7K
Obj. (Time) |
|---|---|---|---|---|
| HGS | 41.2 (5m) | 57.2 (5m) | 126.2 (5m) | 172.1 (5m) |
| LKH3 | 46.4 (6.2s) | 64.9 (20s) | 175.7 (2.5m) | 245.0 (8.4m) |
| Random Insertion | 66.3 (<1s) | 95.3 (<1s) | 225.4 (<1s) | 309.2 (<1s) |
| TAM-LKH3 | 46.3 (1.8s) | 64.8 (5.6s) | 144.6 (17s) | 196.9 (33s) |
| TAM-HGS | — | — | 142.8 (30s) | 193.6 (52s) |
| Ours greedy | **44.1** (0.2s) | **59.6** (0.8s) | **131.0** (4.7s) | **176.4** (8.9s) |

## H  RESULTS ON SMALL SCALE INSTANCES

We empirically evaluate our method on small-scale TSP and CVRP (C=100 for all scales) instances and present the results in Table 11. From the results, we can observe that our method consistently demonstrates outstanding performance. For both TSP and CVRP, our method with PRC outperforms the other learning-based methods in terms of solution quality, while maintaining a shorter or equal inference time in the majority of instances except for TSP300. Although slightly more time is required for the TSP300 instance, this is accompanied by a notable improvement in solution quality. These findings highlight the effectiveness and efficiency of our method.

Table 11: Comparative results on small-scale TSP and CVRP instances.

| Method | TSP300
Obj. | Gap | Total Time | TSP500
Obj. | Gap | Total Time | TSP700
Obj. | Gap | Total Time |
|---|---|---|---|---|---|---|---|---|---|
| Concorde | 12.973 | 0.000% | 5.9m | 16.522 | 0.000% | 32m | 19.442 | 0.000% | 1.8h |
| H-TSP | 13.537 | 4.346% | 15.8s | 17.549 | 6.219% | 23s | 20.694 | 6.444% | 29.1s |
| GLOP | 13.137 | 1.263% | 1.4m | 16.883 | 2.186% | 1.6m | 19.963 | 2.683% | 3.0m |
| POMO aug×8 | 13.834 | 6.636% | 16.8s | 20.187 | 22.189% | 1.1m | 25.576 | 31.555% | 3.7m |
| INViT-3V greedy | 13.725 | 5.797% | 4.8m | 17.327 | 4.877% | 11.7m | 20.712 | 6.536% | 12.2m |
| LEHD greedy | 13.117 | 1.104% | 4.8s | 16.780 | 1.560% | 16s | 19.848 | 2.088% | 37s |
| BQ greedy | 13.071 | 0.752% | 19.3s | 16.717 | 1.180% | 46s | 19.752 | 1.597% | 1.2m |
| SIGD greedy | 13.080 | 0.822% | 11s | 16.714 | 1.166% | 29s | 19.692 | 1.290% | 59s |
| Ours | **13.063** | **0.695%** | 13.9s | **16.700** | **1.083%** | 16s | **19.678** | **1.216%** | 36s |

| Method | CVRP300
Obj. | Gap | Total Time | CVRP500
Obj. | Gap | Total Time | CVRP700
Obj. | Gap | Total Time |
|---|---|---|---|---|---|---|---|---|---|
| HGS | 23.978 | 0.000% | 4.2h | 36.561 | 0.000% | 4h | 49.836 | 0.000% | 4.2h |
| POMO aug×8 | 26.985 | 12.538% | 18s | 44.638 | 22.091% | 1.2m | 73.560 | 47.603% | 3.4m |
| LEHD greedy | 25.182 | 5.020% | 5s | 38.413 | 5.064% | 17s | 50.930 | 2.194% | 37s |
| INViT-3V greedy | 26.873 | 12.070% | 6.1m | 41.133 | 12.505% | 10.4m | 54.838 | 10.036% | 15.4m |
| BQ greedy | 25.155 | 4.906% | 26.9s | 38.438 | 5.134% | 47s | 51.020 | 2.375% | 1.4m |
| SIGD greedy | 25.148 | 4.878% | 22s | 38.671 | 5.770% | 42s | 51.656 | 3.651% | 1.6m |
| Ours | **24.961** | **4.099%** | 5s | **37.903** | **3.671%** | 16s | **50.221** | **0.771%** | 32s |

## I  EXPANSIBILITY

We evaluate the expansibility of our method by applying it to another NCO method BQ (Drakulic et al., 2023). Specifically, we replace the self-attention within BQ with the proposed cross-attention, and then train it by SIT on CVRP1K and CVRP5K directly. We then use the greedy search with the default setting to test the model performance. The results displayed in Table 12 demonstrate that the proposed method can significantly boost the performance of BQ on large-scale problems in terms of gap, inference time, and memory usage.

Table 12: Expansibility of our method to BQ (Drakulic et al., 2023). Gap (%), Time (s), and Memory (MB) are averaged over instances.

|  | CVRP1K | | | CVRP5K | | |
|---|---|---|---|---|---|---|
|  | Gap | Time | Memory | Gap | Time | Memory |
| BQ origin | 8.23 | 1.0 | 20.6 | 20.48 | 8.1 | 479.1 |
| BQ+Cross-Attention+SIT | **7.46** | **0.3** | **3.1** | **8.19** | **3.2** | **5.0** |

## J  EXPERIMENTS AND DISCUSSION ON THE BASELINES RETRAINED ON LARGE-SCALE INSTANCES

We did not train baseline NCO models on problem instances as large as those our model uses for the following reasons: 1) For NCO methods that use supervised learning for model training, it is very difficult to obtain sufficient labeled data (i.e., optimal solutions of problem instances with such large sizes). 2) For NCO methods based on reinforcement learning, excessive GPU memory overhead prevents them from using such large-scale problem instances for model training.

We agree that training the model with larger problem instances can increase its performance on large-scale problems. Herein, we train two representative models (i.e., POMO and LEHD) using instances with problem sizes as large as possible. For POMO, considering the GPU memory limit, we use TSP500 to perform reinforcement learning-based model training. For LEHD, we use LKH3 to get the labels (i.e., optimal solutions) for TSP1K instances, and conduct its supervised learning-based model training. The two models are compared with ours on in-domain data (the same size as the training data) and out-of-domain data (larger than the training data size), and the results are provided in Table 13 and Table 14. These results reveal that our method significantly outperforms the two baseline models on both in-domain and out-of-domain data.

We would like to emphasize that the main contribution of this work precisely lies in overcoming the limitations of reinforcement learning and supervised learning methods for NCO, allowing the model to be directly trained with problem instances whose sizes are larger than 1K. This innovation is key to enabling the NCO method to solve large-scale problems effectively.

Table 13: Comparison on in-domain data (the same size as the training data)

|  | TSP500 | |  |  | TSP1K | |
|---|---|---|---|---|---|---|
|  | Gap | Time |  |  | Gap | Time |
| POMO, aug×8 | 4.88% | 0.5s |  | LEHD, greedy | 2.37% | 0.8s |
| Ours, greedy | **2.19%** | **0.1s** |  | Ours, greedy | **1.95%** | **0.2s** |

## K  STUDY ON THE GENERALIZABILITY OF SIT AND THE EFFECTIVENESS OF THE CROSS-ATTENTION MODEL

To illustrate the generalizability of SIT, we apply it to two other models: POMO (Kwon et al., 2020) and LEHD (Luo et al., 2023). The two models are trained via SIT using the dataset of TSP1K and tested on datasets TSP1K, TSP5K, and TSP10K. From the results in Table 15, we can see

Table 14: Comparison on out-of-domain data (larger than the training data size)

|  | TSP5K | | TSP10K | |
| --- | --- | --- | --- | --- |
|  | Gap | Time | Gap | Time |
| POMO trained on TSP500, aug×8 | 56.05% | 8.6m | OOM | |
| LEHD trained on TSP1K, greedy | 5.84% | 1.5m | 12.45% | 11.7m |
| Ours, greedy | **3.84%** | **5.2s** | **4.62%** | **20.1s** |

that POMO+SIT outperforms the original POMO by 70.5% and 61.7% on TSP1K and TSP5K, respectively. Meanwhile, LEHD+SIT improves the original LEHD by 36.3%, 58.6%, and 49.3% on three kinds of problem instances. These results suggest that SIT can be generalized to other NCO models that enable them to be trained on larger problem instances, thereby improving their ability to solve large-scale problems.

Table 15: Generalizability of SIT on POMO and LEHD

|  | TSP1K | | TSP5K | | TSP10K | |
| --- | --- | --- | --- | --- | --- | --- |
|  | Gap | Time | Gap | Time | Gap | Time |
| POMO, aug×8 | 40.6% | 4.1s | 87.72% | 8.6m | OOM | |
| POMO + SIT, aug×8 | **11.96%** | 4.1s | **25.99%** | 8.6m | OOM | |
| LEHD, greedy | 3.11% | 0.8s | 15.46% | 1.5m | 27.24% | 11.7m |
| LEHD + SIT, greedy | **1.98%** | 0.8s | **6.40%** | 1.5m | **13.81%** | 11.7m |

To demonstrate the effectiveness of our cross-attention model, we compare it with LEHD (Luo et al., 2023) under the same SIT training setting (i.e., trained using TSP1K and tested on TSP1K, TSP5K, and TSP10K). The experimental results in Table 16 indicate that our model significantly outperforms LEHD in all three aspects. Our model achieves 0.03%, 2.56%, and 9.19% optimality gap improvement over LEHD. In addition, its inference speed is 4×, 17×, and 34× that of LEHD, and its memory usage is 9.9%, 2.1%, and 1.02% of LEHD.

Table 16: Comparison between our model and LEHD

|  | TSP1K | | | TSP5K | | | TSP10K | | |
| --- | --- | --- | --- | --- | --- | --- | --- | --- | --- |
|  | Gap | Time | Memory | Gap | Time | Memory | Gap | Time | Memory |
| LEHD + SIT, greedy | 1.98% | 0.8s | 97.4MB | 6.40% | 1.5m | 2323.2MB | 13.81% | 11.7m | 9224.3MB |
| Our model + SIT, greedy | **1.95%** | **0.2s** | **9.65MB** | **3.84%** | **5.2s** | **47.94MB** | **4.62%** | **20.1s** | **94.41MB** |

## L   COMPARISON WITH NEUROLKH

We additionally include the experimental results of NeuroLKH (Xin et al., 2021b) for comparison in Table 17 and Table 18. The results show that NeuroLKH obtains optimal or near-optimal solutions on TSP1K, TSP5K, and TSP10K. However, it fails to solve TSP50K, and TSP100K due to singular values or the out-of-memory issue. In contrast, our model can achieve high-quality solutions for problem instances of all five sizes. Moreover, NeuroLKH is significantly worse than our model on CVRP instances. This is because NeuroLKH relies on heuristic solver LKH3, which is designed for TSP and is less efficient in solving CVRP.

Table 17: Comparion results with NeuroLKH on uniformly sampled TSP and CVRP instances

| Method | TSP1K Gap | TSP1K Time | TSP5K Gap | TSP5K Time | TSP10K Gap | TSP10K Time | TSP50K Gap | TSP50K Time | TSP100K Gap | TSP100K Time |
|---|---|---|---|---|---|---|---|---|---|---|
| NeuroLKH (1000 trials) | **0.00%** | 27s | **0.003%** | 3.7m | **0.0069%** | 8.9m | N/A | | N/A | |
| Ours PRC100 | 0.58% | 9.4s | 1.67% | 52.0s | 2.11% | 1.7m | **2.91%** | 8.6m | **2.90%** | 17m |

| Method | CVRP1K Gap | CVRP1K Time | CVRP5K Gap | CVRP5K Time | CVRP10K Gap | CVRP10K Time | CVRP50K Gap | CVRP50K Time | CVRP100K Gap | CVRP100K Time |
|---|---|---|---|---|---|---|---|---|---|---|
| NeuroLKH (1000 trials) | 4.20% | 15.6s | 19.65% | 3.6m | 41.52% | 14.1m | N/A | | N/A | |
| Ours PRC100 | **3.31%** | 8.0s | **2.05%** | 46s | **0.59%** | 1.3m | **0.11%** | 5.49m | **-1.00%** | 11.5m |

Table 18: Comparion results with NeuroLKH on TSPLIB and CVRPLIB. $^\dagger$: the method encounters problems in solving one or more problem instances

| Method | TSPLIB 1K < n ≤ 5K | TSPLIB n > 5K | TSPLIB All | TSPLIB Solved # | CVRPLIB 1K < n ≤ 7K | CVRPLIB n > 7K | CVRPLIB All | CVRPLIB Solved # |
|---|---|---|---|---|---|---|---|---|
| NeuroLKH (1000 trials) | 0.22% | 0.05%$^\dagger$ | 0.20% | 27/33 | 11.58% | 41.91%$^\dagger$ | 21.70% | 12/14 |
| Ours PRC1,000 | **1.576%** | **4.043%** | **2.556%** | 33/33 | **8.347%** | **11.209%** | **9.574%** | 14/14 |

## M EFFECT OF AUGMENTATION METHOD

We include the experimental results of our model with data augmentation (i.e., aug×8 (Kwon et al., 2020)) in Table 19. These results indicate that the aug×8 improves the model's accuracy on each kind of TSP or CVRP instance, but it also consumes more inference time. Specifically, the optimality gap is improved by 0.73% on average, but the inference time increases by an average of 7.1 times.

Table 19: Comparative results on synthetic TSP and CVRP instances.

| Method | TSP1K Obj. (Gap) | TSP1K Time | TSP5K Obj. (Gap) | TSP5K Time | TSP10K Obj. (Gap) | TSP10K Time |
|---|---|---|---|---|---|---|
| Ours greedy | 23.569 (1.95%) | 0.2s | 52.59 (3.17%) | 5.2s | 74.69 (4.05%) | 20.1s |
| Ours greedy + aug×8 | 23.391 (1.18%) | 1.5s | 52.11 (2.22%) | 36s | 73.99 (3.08%) | 2.3m |
| Ours PRC50 | 23.279 (0.69%) | 4.6s | 51.92 (1.85%) | 23.4s | 73.41 (2.27%) | 49.0s |
| Ours PRC50 + aug×8 | 23.22 (0.45%) | 32.5s | 51.82 (1.66%) | 2.6m | 73.29 (2.10%) | 6.4m |

| Method | CVRP1K Obj. (Gap) | CVRP1K Time | CVRP5K Obj. (Gap) | CVRP5K Time | CVRP10K Obj. (Gap) | CVRP10K Time |
|---|---|---|---|---|---|---|
| Ours greedy | 38.11 (5.01%) | 0.2s | 92.44 (3.01%) | 5.5s | 109.02 (1.50%) | 20.6s |
| Ours greedy + aug×8 | 37.74 (3.99%) | 1.5s | 91.86 (2.37%) | 37s | 107.96 (0.52%) | 2.4m |
| Ours PRC50 | 37.57 (3.54%) | 3.5s | 92.06 (2.58%) | 19.9s | 108.79 (1.29%) | 34s |
| Ours PRC50 + aug×8 | 37.35 (2.92%) | 24.7s | 91.36 (1.80%) | 2.2m | 107.28 (-0.11%) | 4.0m |

# N    EFFECT OF MODEL SIZE

The sizes of all compared constructive models are summarized in Table 20. Our model size is 2.63M, which is larger than POMO, LEHD, and SIGD, but smaller than INViT and BQ.

Table 20: Comparion results on TSP instances. OOM: the method exceeded memory limits

| Method | Model Size | TSP1K | | TSP5K | | TSP10K | | TSP50K | | TSP100K | |
| | | Gap | Time | Gap | Time | Gap | Time | Gap | Time | Gap | Time |
|---|---|---|---|---|---|---|---|---|---|---|---|
| POMO aug×8 | 1.27M | 40.6% | 4.1s | 72.1% | 8.6m | OOM | | OOM | | OOM | |
| INViT-3V greedy | 2.64M | 6.66% | 9.0s | 6.90% | 1.2m | 7.07% | 3.7m | 7.18% | 1.3h | 7.20% | 5.0h |
| LEHD greedy | 1.43M | 3.11% | 0.8s | 15.46% | 1.5m | 27.24% | 11.7m | OOM | | OOM | |
| BQ greedy | 3.11M | 2.30% | 0.9s | 14.31% | 22.5s | 25.02% | 1.0m | OOM | | OOM | |
| SIGD greedy | 1.78M | 1.96% | 1.2s | 12.20% | 1.8m | 30.68% | 15.5m | OOM | | OOM | |
| Ours greedy | 2.63M | **1.95%** | **0.2s** | **3.17%** | **5.2s** | **4.05%** | **20.1s** | **5.36%** | **7.7m** | **6.13%** | **33.0m** |

We agree that more parameters may produce better results. However, the increase in model size cannot lead to performance improvement as remarkable as the innovation of model structure and training schemes. To demonstrate it, we increase the model sizes of SIGD and LEHD to 3.11M and 3.01M and retrain them for comparison. Since the performance of POMO is significantly worse than other methods, we do not consider retraining and comparing it. The experimental results of the two retrained models along with INViT (2.64M parameters) and BQ (3.11M parameters) are listed in Table 21. These results reveal that the four state-of-the-art methods are significantly worse than our method, despite that their model sizes are larger than ours.

We believe that our method's good performance in solving large-scale problems benefits a lot from the SIT training strategy that enables our model to be trained directly using large-scale instances. The proposed SIT opens a door for scalable NCO methods and we believe it can inspire more follow-up works on solving large-scale VRPs.

Table 21: Comparion results on TSP instances. Models are tested using greedy search. OOM: the method exceeded memory limits

| Model | TSP1K | | TSP5K | | TSP10K | | TSP50K | | TSP100K | |
| | Gap | Time | Gap | Time | Gap | Time | Gap | Time | Gap | Time |
|---|---|---|---|---|---|---|---|---|---|---|
| BQ (3.11M) | 2.30% | 0.9s | 14.31% | 22.5s | 25.02% | 1.0m | OOM | | OOM | |
| INViT-3V (2.64M) | 6.66% | 9.0s | 6.90% | 1.2m | 7.07% | 3.7m | 7.18% | 1.3h | 7.20% | 5.0h |
| SIGD (3.11M) | 2.69% | 1.8s | 9.68% | 2.9m | 17.84% | 23.2m | OOM | | OOM | |
| LEHD (3.01M) | 2.10% | 1.7s | 9.58% | 3.7m | 18.30% | 27.2m | OOM | | OOM | |
| Ours (2.63M) | **1.95%** | 0.2s | **3.17%** | 5.2s | **4.05%** | 20.1s | **5.36%** | 7.7m | **6.13%** | 33.0m |

## O  COMPARISON WITH RECENTLY PUBLISHED RL-BASED METHODS

In the main text, we already compared several recently published RL-based methods, including GLOP (AAAI 2024) (Ye et al., 2024) and INViT (ICML 2024) (Fang et al., 2024). Beyond that, We herein include two additional RL-based methods ELG (IJCAI 2024) (Gao et al., 2024) and UDC (NeurIPS 2024) (Zheng et al., 2024) for comparison, and the results are presented in Table 22. From the results, we can observe that our method achieves the best performance among all competitors on all scales of problem instances. Specifically, our method with only 10 PRC iterations can beat these RL-based methods in terms of both solution quality and solving efficiency on all scales of TSP instances. For CVRP, our method using the basic greedy search already achieves better performance than the other methods concerning optimality gap and running time.

The advantages of our method can be summarized in the following two aspects. Firstly, the proposed cross-attention model owns a linear complexity, enabling our method to have a much better inference efficiency, especially when dealing with problem instances with sizes larger than 1K. Secondly, the proposed Self-Improved Training (SIT) allows the model to be directly trained using large-scale (even larger than 1K) instances. Using large-scale instances for model training equips the model with a better ability to solve large-scale problems.

A disadvantage of our method is that it requires multiple iterations to obtain high-quality pseudo-labels for self-improved training. An interesting future work to address this issue is to develop more effective schemes to generate high-quality pseudo-labels.

Table 22: Comparion results on synthetic TSP and CVRP instances. OOM: the method exceeded memory limits

| Method | TSP1K Gap | TSP1K Time | TSP5K Gap | TSP5K Time | TSP10K Gap | TSP10K Time | TSP50K Gap | TSP50K Time | TSP100K Gap | TSP100K Time |
|---|---|---|---|---|---|---|---|---|---|---|
| LKH3 | 0.00% | 1.7m | 0.00% | 12m | 0.00% | 33m | 0.00% | 10h | 0.00% | 25h |
| GLOP (AAAI 2024) | 2.85% | 10.2s | 4.26% | 1.0m | 4.39% | 1.9m | 5.10% | 1.5m | 5.14% | 3.9m |
| ELG aug×8 (IJCAI 2024) | 11.33% | 0.8s | 18.08% | 21s | OOM | | OOM | | OOM | |
| INViT-3V greedy (ICML 2024) | 6.66% | 9.0s | 6.90% | 1.2m | 7.07% | 3.7m | 7.18% | 1.3h | 7.20% | 5.0h |
| UDC ($x$=1000) (NeurIPS 2024) | 1.70% | 1.4m | OOM | | OOM | | OOM | | OOM | |
| Ours greedy | 1.95% | 0.2s | 3.17% | 5.2s | 4.05% | 20.1s | 5.36% | 7.7s | 6.13% | 33.0m |
| Ours PRC10 | 1.20% | 0.9s | 2.73% | 5.1s | 3.08% | 10.0s | 4.22% | 1.33m | 4.16% | 3.0m |
| Ours PRC100 | **0.58%** | 9.4s | **1.67%** | 52.0s | **2.11%** | 1.7m | **2.91%** | 8.6m | **2.90%** | 17m |

| Method | CVRP1K Gap | CVRP1K Time | CVRP5K Gap | CVRP5K Time | CVRP10K Gap | CVRP10K Time | CVRP50K Gap | CVRP50K Time | CVRP100K Gap | CVRP100K Time |
|---|---|---|---|---|---|---|---|---|---|---|
| HGS | 0.00% | 2.5m | 0.00% | 2.0h | 0.00% | 5.0h | 0.00% | 8.1h | 0.00% | 24h |
| GLOP-G (LKH3) (AAAI 2024) | 8.83% | 1.3s | 10.2% | 6.8s | 8.27% | 11.2s | OOM | | OOM | |
| ELG aug×8 (IJCAI 2024) | 14.56% | 1.1s | 22.06% | 30s | OOM | | OOM | | OOM | |
| INViT-3V greedy (ICML 2024) | 17.8% | 11.4s | 22.41% | 1.4m | 31.66% | 4.2m | 50.17% | 2.9h | 44.67% | 8.3h |
| UDC ($x$=1000) (NeurIPS 2024) | 5.29% | 6.2m | OOM | | OOM | | OOM | | OOM | |
| Ours greedy | 5.01% | 0.2s | 3.01% | 5.49s | 1.50% | 20.62s | 0.60% | 8.06m | -0.22% | 33.1m |
| Ours PRC10 | 4.52% | 0.7s | 4.65% | 3.9s | 4.43% | 6.8s | 6.52% | 28s | 4.23% | 59s |
| Ours PRC100 | **3.31%** | 8.0s | **2.05%** | 46s | **0.59%** | 1.3m | **0.11%** | 5.49m | **-1.00%** | 11.5m |

## P  EXPERIMENTS ON LARGE-SCALE VRP INSTANCES BEYOND 100K NODES

Our method can scale to VRP instances beyond 100K nodes in a zero-shot manner. To demonstrate this, we use the model trained on TSP100K to solve TSP200K and TSP500K instances. The results in Table 23 indicate that our method can successfully solve these extremely large-scale instances with relatively low optimality gaps.

The model's performance can be further improved if it is fine-trained using larger problem instances (e.g., on TSP500K). However, the time required to generate pseudo-labels increases with the size of the training data. Thereby, training models on very large-scale VRP instances can result in prohibitively long training times. A possible coping strategy is to adopt multi-GPU parallel training, which will be studied in our future work.

Table 23: Results on TSP200K and TSP500K instances

| Scale | TSP200K | | | TSP500K | | |
|---|---|---|---|---|---|---|
| | Gap | Time | Memory | Gap | Time | Memory |
| Ours PRC100 | 3.57% | 33m | 3166MB | 6.03% | 1.1h | 7916MB |

## Q  LICENSES

The licenses for the codes and the datasets used in this work are listed in Table 24.

Table 24: List of licenses for the codes and datasets we used in this work.

| Resource | Type | Link | License |
|---|---|---|---|
| LKH3 (Helsgaun, 2017) | Code | http://webhotel4.ruc.dk/ keld/research/LKH-3/ | Available for academic research use |
| HGS (Vidal, 2022) | Code | https://github.com/chkwon/PyHygese | MIT License |
| Concorde (Applegate et al., 2006) | Code | https://github.com/jvkersch/pyconcorde | BSD 3-Clause License |
| POMO (Kwon et al., 2020) | Code | https://github.com/yd-kwon/POMO | MIT License |
| LEHD (Luo et al., 2023) | Code | https://github.com/CIAM-Group/NCO_code/tree/main/single_objective/LEHD | MIT License |
| BQ (Drakulic et al., 2023) | Code | https://github.com/naver/bq-nco | CC BY-NC-SA 4.0 |
| GLOP (Ye et al., 2024) | Code | https://github.com/henry-yeh/GLOP | MIT License |
| H-TSP (Pan et al., 2023) | Code | https://github.com/Learning4Optimization-HUST/H-TSP | MIT License |
| DIFUSCO (Sun & Yang, 2023) | Code | https://github.com/Edward-Sun/DIFUSCO | MIT License |
| INViT (Fang et al., 2024) | Code | https://github.com/Kasumigaoka-Utaha/INViT | Available for academic research use |
| Att-GCN+MCTS (Fu et al., 2021) | Code | https://github.com/SaneLYX/TSP_Att-GCRN-MCTS | MIT License |
| ELG (Gao et al., 2024) | Code | https://github.com/gaocrr/ELG | MIT License |
| NeuroLKH (Xin et al., 2021b) | Code | https://github.com/liangxinedu/NeuroLKH | Available for academic research use |
| UDC (Zheng et al., 2024) | Code | https://github.com/CIAM-Group/NCO_code/tree/main/single_objective/UDC-Large-scale-CO-master | MIT License |
| SIGD (Pirnay & Grimm, 2024) | Code | https://github.com/grimmlab/gumbeldore | Available for academic research use |
| TSPLIB (Reinelt, 1991) | Dataset | http://comopt.ifi.uni-heidelberg.de/software/TSPLIB95/ | Available for any non-commercial use |
| CVRPLib (Uchoa et al., 2017) | Dataset | http://vrp.galgos.inf.puc-rio.br/index.php/en/ | Available for academic research use |

