# OpenReview forum: "Boosting Neural Combinatorial Optimization for Large-Scale Vehicle Routing Problems"
_ICLR.cc/2025/Conference — ICLR 2025 Poster_

### Official Review · Reviewer_w834 · 2024-10-31

**Soundness:** 3
**Presentation:** 3
**Contribution:** 3
**Rating:** 6
**Confidence:** 2

**Summary:**

The paper addresses the challenge of solving large-scale vehicle routing problems (VRPs) using neural combinatorial optimization (NCO). The authors propose a novel approach that combines a lightweight cross-attention mechanism with a self-improved training (SIT) algorithm. The method is evaluated on multiple benchmark datasets, demonstrating significant improvements over existing methods in terms of solution quality and computational efficiency.

**Strengths:**

1) The paper introduces a lightweight cross-attention mechanism that significantly reduces the computational complexity of NCO models, addressing a key limitation in handling large-scale VRPs.

2) The experimental setup is robust, using a variety of benchmark datasets and comparing against multiple state-of-the-art methods.

3) The paper is well-structured and clearly written, making it easy to follow the methodology and understand the contributions.

**Weaknesses:**

1) Although the paper compares against several state-of-the-art methods, a more detailed comparison with recent advancements in reinforcement learning-based approaches for VRPs could provide a more comprehensive evaluation.

2) An exploration of the limitations of the cross-attention mechanism and potential areas for improvement would enhance the paper's depth.

**Questions:**

1) How does the proposed method scale to even larger problem instances beyond 100K nodes? Are there any known limitations or bottlenecks in the current implementation?

2) How does the proposed method compare to recent reinforcement learning-based approaches for VRPs in terms of solution quality and computational efficiency? Are there any specific advantages or disadvantages?

---

### Official Review · Reviewer_S6Ze · 2024-11-04

**Soundness:** 3
**Presentation:** 3
**Contribution:** 3
**Rating:** 6
**Confidence:** 4

**Summary:**

The paper proposes an efficient cross attention mechanism and a self-improved training (SIT) procedure for scaling up NCO solvers. The paper introduces representative embeddings that serve as an information bottleneck to reduce the complexity of attention. SIT algorithm iteratively improves the solution by using parallel local reconstructions. The resulting model achieves low complexity and is scalable to large VRP instances.

**Strengths:**

- The efficient cross attention mechanism is a novel way to reduce the complexity of transformer based NCO solvers. While the idea is simple, the experiments show that cross attention is really effective for large scale VRP.
- The SIT algorithm is simple yet effective although local self improvement is not exactly a novel thing.
- The paper is well written.

**Weaknesses:**

- The Related work section is pushed to the appendix. I know there is a lot to say about the proposed method, but the related work should be in the main text. The related work is not very well written. To improve readability, please divide the text into subsections, each discussing a different approach to the problem.
- The paper did not compare with strong baselines such as ELG (Gao et al., 2024), GLOP (more revisions) (Ye et al., 2024) although the authors mentioned them in the related work section.
- The paper lacks a comparison with other self improvement methods. Some examples of self improvement approach to NCO are [1]
- The paper did not compare the sizes of the models. More parameters often lead to better results. To ensure a fair comparison, the experiment should be conducted on models of the same size.

[1] Self-Improvement for Neural Combinatorial Optimization: Sample without Replacement, but Improvement. Jonathan Pirnay, Dominik G. Grimm

**Questions:**

Please address the weaknesses mentioned above.

---

### Official Review · Reviewer_vFLE · 2024-11-04

**Soundness:** 3
**Presentation:** 3
**Contribution:** 4
**Rating:** 8
**Confidence:** 4

**Summary:**

In their paper, the authors present a deep learning-based, lightweight model and training scheme for solving VRPs. Specifically, the propose replacing the expensive self-attention mechanism in transformer-based solvers with cross-attention based on the first and last node of the currently constructed route. Additionally, they rely on Self-improved training (SIT) as a training mechanism instead of reinforcement or supervised learning. SIT decomposes the total solution into multiple partial solutions and resolves them to find the optimal subsolution.

**Strengths:**

The paper presents a novel learning method for deep-learning-based approaches in VRPs. SIT seems to improve performance significantly with the cross-attention making previously prohibitive problem sizes now possible. The performance of the model also seems to be very good, consistently beating the baselines.

**Weaknesses:**

I have some concerns about the evaluation procedure. Specifically, current transformer-based methods such as POMO are trained on small-scale problems (around 100 nodes) and are known to generalize poorly to bigger problems. Comparing to pre-trained models on the larger problems is a bit misleading as in this scenario the authors' method is trained and tested on the same size problem while the other baselines are forced to generalize.

Moreover, and especially considering the SIT training scheme, it might have been good to see a replacement where these deep learning methods are applied on the partial problems with a size of 100 etc. that is closer to their training setting. This would help both see the effectiveness of SIT in other models and better compare with the introduced cross-attention. Alternatively, a direct comparison on smaller problems (of the same training size as the baselines) would have been useful.

 Finally, an important baseline would have been NeuroLKH [1] which combines deep learning methods with the LKH algorithm. It was shown to consistently beat LKH with much smaller runtimes which would be interesting to see since, in some cases, the margin between LKH and the authors' methods is not so large.

[1] Xin, Liang, et al. "Neurolkh: Combining deep learning model with lin-kernighan-helsgaun heuristic for solving the traveling salesman problem." Advances in Neural Information Processing Systems 34 (2021): 7472-7483.

**Questions:**

Are the baselines retrained in this scenario?
Have you tried using augmentation methods similar to POMO on your own cross-attention transformer?
How generalizable to other models do you think this SIT method is?

---

### Official Review · Reviewer_R2Zn · 2024-11-08

**Soundness:** 2
**Presentation:** 3
**Contribution:** 2
**Rating:** 5
**Confidence:** 3

**Summary:**

This paper presents a lightweight cross-attention mechanism with linear complexity to improve the efficiency of the NCO model in solving large-scale VRPs. By propagating node embeddings through representative nodes, the cross-attention mechanism maintains effective interactions between nodes while achieveing low complexity. The experimental results on TSP and CVRP with scaling up to 100K show that the proposed method is effective

**Strengths:**

The complexity of the proposed method is lower than the traditional methods.

**Weaknesses:**

The complexity analysis seems to be unfair to compare with the traditional method in that the proposed method should consider not only the computational complexity in single round, but also accumulate all computation in iterative reconstruction.

**Questions:**

Pls refer to the weaknesses.

---

### Meta-Review · Area_Chair_Lqur · 2024-12-19

**Metareview:**

The paper presents a novel learning method for deep-learning-based approaches in VRPs. By propagating node embeddings through representative nodes, the proposed cross-attention mechanism maintains effective interactions between nodes while achieveing low complexity, which improves performance significantly with the cross-attention making previously prohibitive problem sizes now possible.  The experimental setup is robust, using a variety of benchmark datasets and comparing against multiple state-of-the-art methods. And the proposed method consistently beats the baselines. The paper is well-structured and clearly written, making it easy to follow the methodology and understand the contributions.

**Additional Comments On Reviewer Discussion:**

I've read the authors' response, as well as the dicussion between reviewers and authors, I lean to accept this paper.

---

### Decision · Program_Chairs · 2025-01-22

Accept (Poster)